# Unifying Object-Centric World Models and Diffusion Policy: A Hierarchical Framework for Multi-Stage Robotic Tasks

## Abstract

Visual world models have shown great potential in learning complex system dynamics. Recent advancements leverage these models as transition functions within Model Predictive Control (MPC) frameworks to solve various control tasks. When applied to robotics, however, they are limited to single-stage tasks such as reaching or grasping, and struggle with multi-stage ones that demand complex sequential planning. In this work, we introduce WorldDP, a world model framework designed for multi-stage robotic manipulation. Our hierarchical approach utilizes a high-level world model as a transition function to optimize for feasible subgoals during runtime, which are subsequently reached by a low-level Diffusion Policy. To further aid in learning dynamics and planning, we incorporate object-centric representations that decouple environmental entities and enable us to plan sequentially with respect to each. Evaluated across several robotics benchmarks, WorldDP consistently outperforms existing baselines, validating that coupling the world model's physically grounded planning with diffusion policy's efficient execution yields superior multi-stage performance.

## 1 Introduction

World models, in recent times, have emerged as a powerful class of functions for learning complex system dynamics from observations (Hou et al., 2026). Since images are the most prevalent data source, visual world models (Hafner et al., 2025; Zhou et al., 2024; Bar et al., 2025; Goswami et al., 2025) have become the standard for capturing these dynamics. Following prior literature (Zhou et al., 2024; Goswami et al., 2025), we define "dynamics" as how the agent moves in response to actions, how the environment reacts, and how both appear from the camera's viewpoint. While early approaches treated raw pixels as system states, more recent methods often adopt the *Joint Embedding Predictive Architecture (JEPA)* (Assran et al., 2025) paradigm, representing states as latent embeddings of image encoders (Zhou et al., 2024).

World models, when applied to control tasks, typically follow one of three primary methodologies: integrating them into Reinforcement Learning (RL) pipelines (Hafner et al., 2025; Hansen et al., 2024), directly predicting joint action-state sequences (Cen et al., 2025), or embedding them within an optimal control framework (Sobal et al., 2026). In the latter approach, the world model works as a transition function inside a Model Predictive Control (MPC) loop to plan trajectories by optimizing action sequences. Our work focuses on this third category. A key advantage over end-to-end methods like Diffusion Policy (Chi et al., 2025) or Vision-Language-Action models (Kim et al., 2024; Physical Intelligence et al., 2025; Bjorck et al., 2025) is that world models can be trained on reward-free, suboptimal trajectories and generalize to new task configurations without any fine-tuning.

While existing world models have shown promising results on synthetic tasks like PushT (Zhou et al., 2024; Maes et al., 2026a), such environments typically feature small observation spaces or discrete action sets. When applied to large-scale robotic tasks with continuous action spaces, existing methods generally succeed only on short-horizon or single-step objectives, such as reaching or grasping (Goswami et al., 2025; Maes et al., 2026b; Terver et al., 2025; Assran et al., 2025) and fail on long-horizon manipulation tasks that require multi-stage, sequential control. The vast majority of robotic tasks are, in fact, multi-staged; for instance, even a simple "pick up a book" task requires a sequence of reaching, grasping, and lifting.

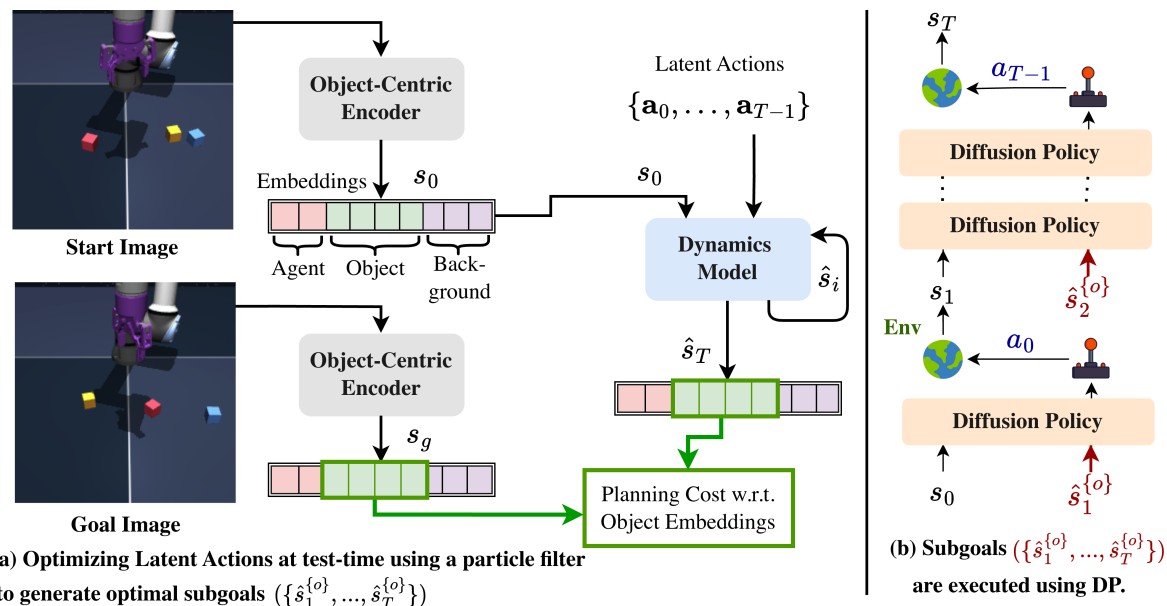

Figure 1: **Overview of WorldDP, a hierarchical framework for multi-stage robotic manipulation.**
(a) At test-time, we use our object-centric world model within a particle filter to optimize latent action sequences, employing a structured, object-based loss to find optimal subgoals. (b) A low-level, goal-conditioned diffusion policy (DP) then sequentially tracks and executes these subgoals to solve the task.

In this paper, we begin by analyzing why current world models struggle with multi-stage tasks and use these insights to motivate our approach. Most existing methods (Zhou et al., 2024; Assran et al., 2025; Terver et al., 2025) utilize patch-level features from large-scale encoders like DINOv2 (Oquab et al., 2023) or V-JEPA (Assran et al., 2025) to represent the system's state space. While effective for computer vision tasks like segmentation, patch-level features are suboptimal system states for dynamics learning as the dynamics models tend to over-focus on dominant agent and background patches, minimizing signals from small objects (e.g., cube) that occupy very few patches. Inspired by Haramati et al. (2026), we adopt an object-centric representation for our world model states. This approach facilitates better attention to relevant entities, such as the agent and environmental items, during model learning and enables individual planning for each item during MPC. Unlike Haramati et al. (2026), we learn these representations on top of DINOv2 (Oquab et al., 2023) patch features, using a SAM2 (Ravi et al., 2025) model to provide guidance during training.

Existing world-model-based planners often rely on a single-tier hierarchy: given a goal image, these methods (Maes et al., 2026b; Zhou et al., 2024) optimize the entire action sequence directly. Despite using environmental feedback at each MPC step, this flat planning remains inherently difficult for multi-stage tasks. While recent work (Zhang et al., 2026) explored hierarchical latent world models by optimizing for intermediate "subgoals", the lower level requires direct physical action optimization, which can be sensitive to subgoal selection, especially for robotic tasks. In our framework, we also utilize a hierarchical structure to optimize for subgoals, but instead of using another world model at the lower level, we use a diffusion policy (Chi et al., 2025). This diffusion policy provides several advantages: it is robust for the short-horizon control needed to reach subgoals, executes much faster than costly world-model planning, and can compensate for suboptimal subgoals to a good extent. Intuitively, our world model decomposes multi-stage tasks into reachable subgoals for the diffusion policy. Further, the object-centric representation enables individual, recursive planning for each object.

Another failure mode for current models is the generation of imprecise subgoals. For tasks like "pick and place," or "open drawer," subgoals should ideally represent pivotal frames, such as the moment a handle is gripped. To encourage the generation of such critical states, we train a contact predictor that signals when the robot interacts with an object, incorporating this signal into our MPC cost function. Finally, while most world-model-based planning relies on optimization techniques like the Cross-Entropy Method (CEM), we find that multi-modal optimization via a Particle Filter is superior, as robotic tasks are inherently multi-

modal (i.e., multiple distinct, equally valid action sequences can achieve the same goal). We evaluate our approach on several multi-stage tasks from the OGBench (Park et al., 2025) benchmark, demonstrating that it outperforms existing world models, diffusion policies, and diffusion-based subgoal generation baselines. In summary, our contributions are:

- We introduce WorldDP (Fig. 1), a novel hierarchical framework that integrates object-centric world models with diffusion policies to solve complex, multi-stage robotic tasks.

- We present an object-centric state representation for latent world models that leverages DINOv2 patch-level features guided by SAM2-generated segmentation masks during training.

- We present the first world model approach to employ object-aware planning, incorporating a contact predictor and particle filter optimization to enhance subgoal generation.

- We provide extensive evaluations across multi-stage benchmarks, demonstrating that WorldDP outperforms existing state-of-the-art methods.

## 2 Related Works

**World Models for Robot Learning.** The concept of world models has a long history in control theory (Bryson & Ho, 1969). It also has roots in biological systems, where animals are believed to maintain internal models of their environment (Wolpert et al., 1995; 1998; Miall & Wolpert, 1996). Recently, learning-based methods have brought world models to the forefront by enabling the learning of complex system dynamics from data (LeCun, 2022; Ha & Schmidhuber, 2018). In robotics, comprehensively reviewed in the survey by Hou et al. (2026), world models generally follow one of three paradigms: acting as simulators to train reinforcement learning policies (e.g., Hafner et al. (2025); Hansen et al. (2024)), directly predicting joint action-state sequences (e.g., Cen et al. (2025); Ye et al. (2026)), or generating future states for high-level planning (e.g., Assran et al. (2025); Zhou et al. (2024)). Our work aligns with the third category, focusing on learning from reward-free, unannotated trajectories of observation-action pairs. Specifically, we build upon the Joint Embedding Predictive Architecture (JEPA) framework (Assran et al., 2023; Bardes et al., 2024; Assran et al., 2025), where world models operate over latent embeddings rather than raw pixels. Dino-WM (Zhou et al., 2024), a pioneer work in JEPA-style world models, trained a predictor over frozen vision embeddings (Oquab et al., 2023) and optimizing actions online via MPC and the Cross-Entropy Method. While subsequent works have extended this approach to navigation (Bar et al., 2025), dexterous manipulation (Goswami et al., 2025), and whole-body control (Bai et al., 2026), or focused on representation learning (Maes et al., 2026b) and parallel planning (Psenka et al., 2026), most remain limited to single-stage tasks like grasping or reaching. In contrast, WorldDP extends these capabilities by integrating object-centric world model planing with diffusion policies, enabling sequential, multi-stage robotic manipulation.

**Hierarchical Task Execution.** A natural approach to solving multi-stage tasks is through hierarchical execution, where upper-level hierarchies predict subgoals at different levels of abstraction, while lower-level components track and execute them. This paradigm is widely used in hierarchical optimal control (Falcone et al., 2008; Li et al., 2021; Fang et al., 2019) and reinforcement learning (RL) (Chen et al., 2024; Choi et al., 2026). For instance, HECRL (Haramati et al., 2026) combines diffusion-based subgoal generation with low-level RL tracking, utilizing object-centric representations to improve execution. However, hierarchical planning within latent world models remains largely unexplored. While V-JEPA2 (Assran et al., 2025) relies on manually defined subgoals for pick-and-place tasks, HWM (Zhang et al., 2026) is the first latent world model to optimize latent actions against a patch-level cost function for two-level hierarchical planning, using upper-level for subgoal generation and the lower-level world model to find physical actions to reach the subgoals. In contrast, WorldDP's hierarchical framework optimizes using a object-centric cost function and tracks the resulting subgoals using a diffusion policy. This strategy provides faster, more robust execution, enabling longer multi-stage sequences such as multi-object pick-and-place tasks.

**Object-Centric Learning.** Existing latent world models typically rely on patch-level representations (e.g., Zhou et al. (2024)), where dynamics models tend to over-focus on dominant environment patches,

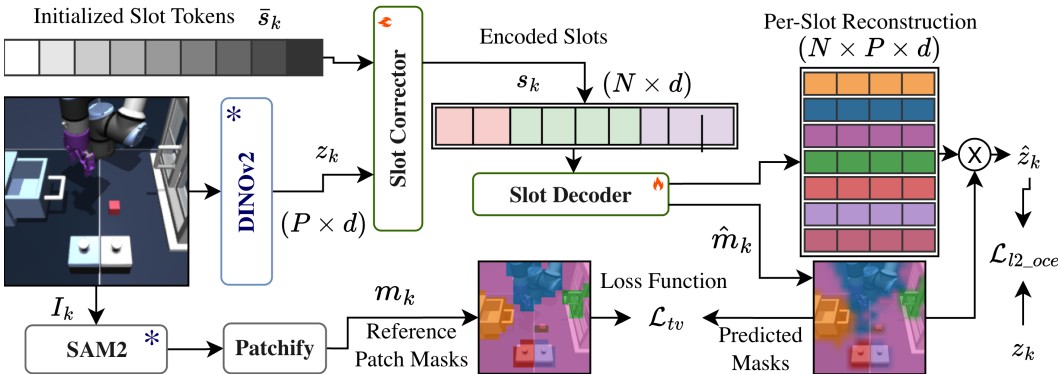

Figure 2: **Object-Centric Encoder Training with SAM2 Guidance.** The DINOv2-encoded image patches $z_k$ and initialized slots $\bar{s}_k$ are refined by a Slot Corrector into $s_k$. A Slot Decoder maps $s_k$ to predicted masks $\hat{m}_k$ and per-slot reconstructions, which reconstruct $\hat{z}_k$. The model is trained using a reconstruction loss (between $z_k$ and $\hat{z}_k$) and a mask segmentation loss against SAM2-generated ground truth $m_k$.

minimizing signals from small objects that occupy very few patches. In contrast, object-centric (or entity-centric) approaches learn distinct embeddings for individual objects. These methods generally fall into two categories: particle-based (Daniel & Tamar, 2022; Qi et al., 2025) and slot-attention-based (Wu et al., 2022; Collu et al., 2024). Particle-based methods decompose an image into a set of particles defined by their spatial locations and local features, a paradigm recently adopted by world models and RL frameworks like Daniel et al. (2026); Haramati et al. (2026). Inspired by Singh et al. (2025); Mosbach et al. (2024), we instead build upon slot attention to extract object-centric representations directly from frozen DINOv2 Oquab et al. (2023) patch features. To enhance feature grounding, we guide slot-attention training with an auxiliary loss derived from SAM2 (Ravi et al., 2025) segmentation masks. Moreover, while existing object-centric world models (e.g., Ferraro et al. (2025)) either predict actions directly or rely on RL, WorldDP is the first to leverage object-centric world models as transition functions within an MPC framework to optimize actions.

## 3 Methodology

**Problem Formulation.** Let $I_k \in \mathbb{R}^{H \times W \times 3}$ ($H$: height, $W$: width) denote the RGB observation of an environment at timestep $k$, captured by a static external camera. The environment contains an articulated robot (e.g., a UR5e arm) and various interactable objects depending on the specific task. Given a current observation $I_0$ and a target goal $I_g$, our objective is to find a sequence of actions $\{a_0, a_1, \dots\}$ to reach the state depicted in $I_g$. The action space $a_k \in \mathbb{R}^5$ consists of commanded changes to the end-effector's $(x, y, z)$ coordinates, yaw, and gripper opening. For training, a "play" dataset of image-action trajectories $(I_0, a_0, I_1, a_1, \dots)$ consisting of robot-environment interactions is available. The dataset lacks explicit reward signals or goal labels, and may contain suboptimal trajectories, including failed manipulation attempts. Our work focuses on multi-stage tasks (e.g., rearrangement of three cubes) which often necessitates high-level planning and sequential manipulation of the various environment entities.

**Method Overview.** The input image $I_k$ is processed by a frozen DINOv2 (Oquab et al., 2023) encoder to extract patch-level features, which our Object-Centric Encoder (Sec. 3.1) aggregates into vector embeddings for each environment entity. These latent embeddings are fed into a Conditional Diffusion Transformer dynamics prediction model (Bar et al., 2025) to predict the subsequent entity states at a future timestep (Sec. 3.2). During task execution, a hierarchical approach is used where the world model serves as a learned transition dynamics function within an MPC framework to identify optimal intermediate subgoals (Sec. 3.3), which are realized via a goal-conditioned Diffusion Policy (Sec. 3.4). All hyperparameters are in the appendix.

### 3.1 Object-Centric Encoder (OCE)

The image $I_k$ is processed by a frozen DINOv2 encoder to generate patch-level features $z_k \in \mathbb{R}^{P \times d}$, where $P$ denotes the number of patches and $d$ the feature dimensionality. Following the slot attention framework (Lo-

catello et al., 2020), we initialize $N$ slots $\bar{s}_k \in \mathbb{R}^{N \times d}$, representing environment entities such as the robot, objects, and background. These slots are refined by a Slot Corrector through recursive passes of cross-attention and Gated Recurrent Unit (GRU) blocks; here, queries and values for attention are derived from $z_k$, while keys are derived from $\bar{s}_k$ using linear layers. The resulting object-centric representation $s_k \in \mathbb{R}^{N \times d}$ serves as the state representation for our world model. To train the Slot Corrector, a Slot Decoder is used, which takes $s_k$ as input and predicts per-slot reconstructions $\hat{z}_k^n \in \mathbb{R}^{P \times d}$ and masks $\hat{m}_k \in \mathbb{R}^{N \times P}$, where $n \in \{1, \ldots, N\}$. Each patch in $\hat{m}_k$ represents a probability distribution over the $N$ entities for that patch. The reconstructed patch-level embedding is calculated as $\hat{z}_k = \sum_{n=1}^{N} \hat{m}_k^n \cdot \hat{z}_k^n$, with the reconstruction loss:

$$\mathcal{L}_{l2\_oce} = \|z_k - \hat{z}_k\|_2^2. \tag{1}$$

Our approach differs from works like Haramati et al. (2026) because we learn the object-centric representation on top of the DINOv2 patch features rather than raw pixels. While this leverages the pre-trained capabilities of DINOv2 and accelerates training, we observe that small objects (e.g., cubes) occupy a small number of patches and often get merged into the robot or background during object-centric learning. We address this by providing privileged guidance during training via SAM2 (Ravi et al., 2025). Images are segmented into entities (robot, background, and specific objects) to generate ground-truth masks. These masks are downsampled to the patch resolution to produce $m_k$; a patch is assigned to an object if it covers more than 15% of the patch area. We employ a Tversky loss (Salehi et al., 2017), $\mathcal{L}_{tv}$, (described in the appendix) to supervise the predicted masks $\hat{m}_k$ (Fig. 2). The total loss for the object-centric encoder (with $\lambda_{oce} = 1$) is:

$$\mathcal{L}_{oce} = \mathcal{L}_{l2\_oce} + \lambda_{oce} \mathcal{L}_{tv}(m_k, \hat{m}_k). \tag{2}$$

## 3.2 Conditional Diffusion Transformer Dynamics Prediction Model

Given the object-centric states $s_{k_1}$ and an action sequence at timestep $k_1$, we next define our dynamics model $f_\theta$. [1] We adopt the Conditional Diffusion Transformer (CDiT) architecture used in Goswami et al. (2025). However, unlike Goswami et al. (2025), which operates on patch-level states of shape $(P, d)$, our dynamics model takes states $s_k \in \mathbb{R}^{N \times d}$ as input to explicitly process the object-centric representation (Fig. 3). Further, the action sequence $a_{k_1 \to k_2} = \{a_{k_1}, a_{k_1+1} \ldots, a_{k_2-1}\}$ (where $k_1 < k_2$) is compressed into a latent action embedding $\mathbf{a}_{k_1} \in \mathbb{R}^{32}$ via a transformer action encoder (Zhang et al., 2026), which conditions the CDiT model through Adaptive Layer Normalization layers.

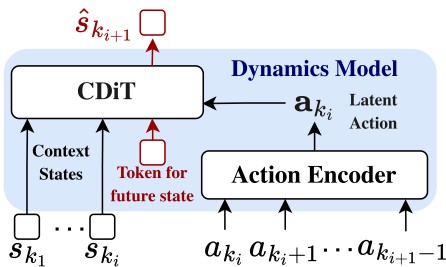

Figure 3: WorldDP's dynamics model.

The object-centric encoder is fully trained beforehand and remains frozen during dynamics model training. For each sample of training, we consider a data horizon of 100 frames, from which we randomly sample 5 frames at timesteps $k_1 < k_2 < k_3 < k_4 < k_5$. While constant frameskips can also be used, utilizing random temporal skips during training has been shown to improve dynamics model robustness (Bai et al., 2026). During training, we predict the next state $\hat{s}_{k_{i+1}}$ conditioned on all previous ground-truth states $\{s_{k_i}, \ldots, s_{k_1}\}$ via teacher forcing, minimized by a mean squared error (MSE) loss. For inference and planning, the model operates autoregressively: the predicted state $\hat{s}_{k_i}$ and the subsequent action chunk $a_{k_i \to k_{i+1}}$ are iteratively fed back into $f_\theta$ to generate $\hat{s}_{k_{i+1}}$, continuing this to roll out future states.

## 3.3 Task Execution

With the dynamics model trained, we formulate our task execution within a hierarchical two-tier MPC framework. The upper tier utilizes the world model to plan for high-level subgoals from a target goal image, while the lower tier employs a goal-conditioned Diffusion Policy to realize these subgoals (Sec. 3.4).

**Sampling-Based Optimization via Particle Filtering.** Starting from the initial state $s_0$, planning involves executing multi-step autoregressive rollouts using $f_\theta$ over a horizon of $T$ world-model steps. The

---

[1]We use the notation $\{k_1, k_2, \ldots\}$ exclusively during training to indicate variable time steps. Elsewhere (e.g., during planning) the notation $\{k, k+1, \ldots\}$ is used for simplicity, as the time steps spacing does not affect the underlying logic.

Algorithm 1: **Pseudo-code of WorldDP's hierarchical planning.** Given the current observation and target goal, a particle filter optimizes latent action sequences through the world model to generate optimal subgoals across sequential object goals. These subgoals are executed via the low-level diffusion policy.

```python
def WorldDPPlanning(obs, global_goal, action_means, **kwargs):
    """
    obs: environment observation
    global_goal: overall task goal image
    action_means: initialized means for the particle filter
    kwargs: contains planning parameters like total_iter, lambda_plan
    """
    for obj_idx in range(num_objects): # num_objects = num_slots − 2
        # Generate sequential goal for current object based on latest observation
        goal = find_curr_object_goal_slot(obs, global_goal, obj_idx)

        # 1. World Model Subgoal Planning via Particle Filtering
        for iter_idx in range(total_iters):
            # A. Generate action particles around the current means
            action_particles = sample_around_mean(action_means, sigma, total_samples)

            # B. Evaluate particles through autoregressive rollouts
            total_costs = []
            for action in action_particles:
                pred_states, object_cost, contact_cost = get_costs(obs, action, goal)
                total_costs.append(object_cost + lambda_plan*contact_cost)

            # C. Select top−M candidates as means for next iteration
            top_M_idx = top_M_indices(total_costs, top_M)
            action_means = [action_particles[i] for i in top_M_idx]

        # 2. Extract optimal subgoal sequence from highest−ranking particle
        optimal_subgoals, _, _ = get_costs(obs, action_means[0], goal)

        # 3. Goal−Conditioned Diffusion Policy to reach subgoals
        curr_obs = env.get_info()
        for subgoal in optimal_subgoals:
            low_level_actions = diff_policy(curr_obs, subgoal)
            curr_obs, is_success = env.step(low_level_actions)

        # Update current observation for the next object planning step
        obs = curr_obs

    return is_success
```

final predicted state $\hat{s}_T$ is evaluated against the goal state $s_g$ (derived from image $I_g$) via a cost function to optimize the latent action sequence. While prior works (Zhou et al., 2024; Bar et al., 2025) use CEM, we employ a Particle Filter (PF) to better handle the multimodal nature of robotic action spaces, where multiple distinct, equally valid action sequences can achieve the same goal.. Unlike CEM, which relies on a single Gaussian distribution, a PF maintains a diverse set of candidate particles to better capture these varied solutions. Notably, WorldDP is the first in this line of work to use a PF for action optimization.

To initialize the PF, we generate spatial trajectories of length $T$ from the end-effector to keypoints, sampled to cover the workspace. These positional deltas are concatenated with randomly initialized yaw and gripper commands, and passed through the action encoder to yield latent seed action sequences. These seeds serve as the initial particles (means) of our filter. We sample $Q$ total action sequences $\bar{\mathbf{a}}^{\{q\}} = \{\mathbf{a}_0^{\{q\}}, \ldots, \mathbf{a}_{T-1}^{\{q\}}\}$ ($q \in \{1, \ldots, Q\}$) around these seed means using a standard deviation $\sigma$. Autoregressive rollouts through

$f_\theta$ generate predicted state trajectories $\mathbf{s}^{\{q\}} = \{s_1^{\{q\}}, \ldots, s_T^{\{q\}}\}$. A cost function $\mathcal{C}(\hat{s}_T^{\{q\}}, s_g)$ evaluates each sequence; the top $M$ candidates are selected as the refined means for the next iteration. This optimization repeats for $L$ iterations, and the highest-ranking sequence at the final step defines the optimal subgoal sequence $\{\hat{s}_1^{\{o\}}, \ldots, \hat{s}_T^{\{o\}}\}$. All the parameters used for the PF for each of the tasks are in the Appendix.

**Object-Centric Cost Function.** As the objective involves manipulating physical objects, our cost function operates directly on the object embeddings. Excluding the agent and background, we isolate the $N-2$ objects embeddings from the predicted state $\hat{s}_T^{\{q\}}$ and denote that as $\hat{o}_T^{\{q\}} \in \mathbb{R}^{(N-2)\times d}$. The first component of our cost is the mean squared error (MSE) relative to the target object goal states $o_g$:

$$\mathcal{C}_{dist} = \frac{1}{(N-2)*d}\|o_g - \hat{o}_T^{\{q\}}\|_2^2. \tag{3}$$

To encourage subgoals to capture critical manipulation phases, such as grasping objects, we incorporate a contact-prediction cost. We train a lightweight MLP contact predictor that maps state $s_k$ to an $(N-2)$-dimensional vector of contact probabilities. Passing the full rollout $\mathbf{s}^{\{q\}}$ yields a contact trajectory $\mathbf{c}^{\{q\}} \in \mathbb{R}^{N-2}$. Given a target object to manipulate, we define a one-hot reference vector $\mathbf{c_{ref}}$ and penalize deviations using a cross-entropy loss $\mathcal{C}_{ce}(\mathbf{c}^{\{q\}}, \mathbf{c_{ref}})$. The final optimization cost is formulated as:

$$\mathcal{C}(\hat{s}_T^{\{q\}}, s_g) = \mathcal{C}_{dist} + \lambda_{plan} \cdot \mathcal{C}_{ce}(\mathbf{c}^{\{q\}}, \mathbf{c_{ref}}) \tag{4}$$

where $\lambda_{plan}$ balances the two objectives. For tasks where only one object changes state between the initial and target configurations (e.g., Scene-Single-Direct in Sec. 4.1), we identify the target using our object-centric encoder. We isolate the object with the largest embedding delta between the start and goal frames, then restrict cost function optimization to that specific object.

**Hierarchical Execution.** The optimal subgoals $\{\hat{s}_1^{\{o\}}, \ldots, \hat{s}_T^{\{o\}}\}$ are tracked sequentially by the low-level Diffusion Policy. A pseudo-code of the planning algorithm is in Alg. 1. For multi-stage tasks requiring sequential manipulation (e.g., Cube-Triple task in Sec. 4.1), we plan and execute all subgoals for one object, capture a new observation, and re-plan for the another object, and so on, maintaining an MPC loop.

## 3.4 Goal-Conditioned Diffusion Policy

We adapt the transformer-based Diffusion Policy (DP) from Chi et al. (2025) by substituting raw DINOv2 inputs with our object-centric representations flattened to a vector of shape $N \times d$. The model is conditioned on a context vector comprising the flattened current and goal states, along with robot end-effector position and velocity, to generate 40-step action sequences. To enhance temporal generalization, we use a randomized goal-sampling strategy that dynamically selects target frames within a 40-frame window, with ground-truth actions zero-padded to length 40. During evaluation, these predicted actions operate within the end-effector's operational space and are executed on the manipulator via an inverse kinematics controller. Furthermore, to ensure precise grasping in pick-and-place tasks, we follow Chang & Gupta (2023) and utilize an end-effector-mounted depth camera to dynamically correct residual pose errors.

## 4 Experiments

Our experiments address the following key questions: (a) How does WorldDP perform in multi-stage robotics tasks compared to existing methods? (b) How does unifying the world model's planning with a diffusion policy in WorldDP impact performance? (c) What is the impact of the object-centric representation? (d) Which framework components are crucial to its success? (e) How does WorldDP generalize to unseen environment conditions like object color, camera view, and lightning?

### 4.1 Implementation Details

**Robotic Tasks and Datasets.** We evaluate our method on manipulation tasks from the OGBench (Park et al., 2025) benchmark, utilizing the environment variants introduced by Maes et al. (2026a). These variants differ from the original benchmark in camera viewpoint and end-effector color. The tasks (Fig. 4) are:

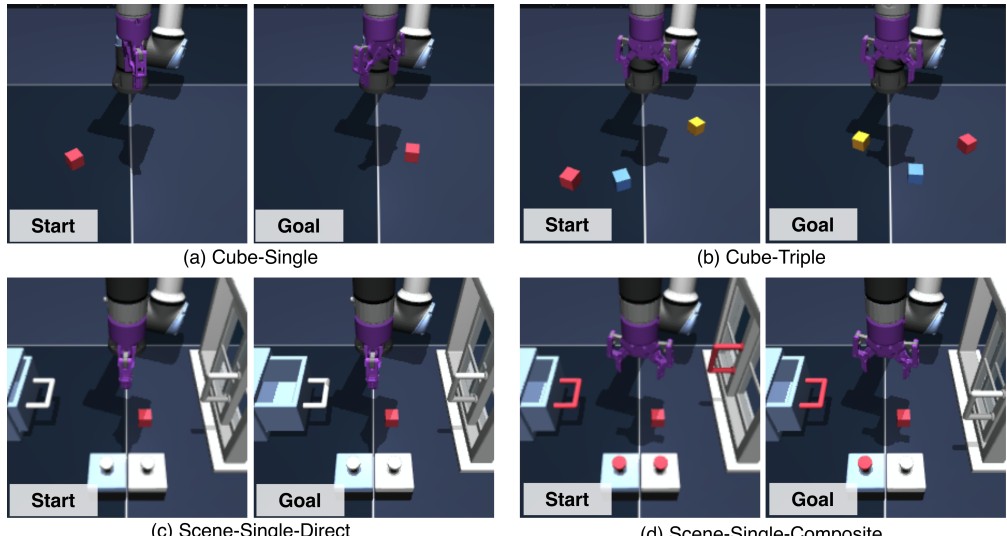

Figure 4: **Example Start and Goal images** from each of the robotic tasks are shown.

(a) **Cube-Single:** A table-top environment featuring a UR5e robotic arm and a single red cube. The robot must manipulate the cube from the initial to a target locations specified by a goal image.

(b) **Cube-Triple:** An expansion of Cube-Single featuring three distinct cubes (red, blue, and yellow). The agent must rearrange them to match a randomly sampled goal configuration.

(c) **Scene-Single-Direct:** This task is based on the Scene-Single environment which includes the UR5e arm, a drawer, a sliding window, a cube, and two push buttons that control the locks for the drawer and window. The goal image differs from the initial state by exactly one object modification (e.g., a single button press or a repositioned window or drawer).

(d) **Scene-Single-Composite:** Utilizing the same environment layout as Scene-Single-Direct, to achieve the goal state, the agent must first interact with the appropriate button to unlock the drawer or window before executing the target manipulation.

Each environment introduces distinct long-horizon and control challenges. While Cube-Single is our baseline task, it still requires precise dynamics and robust control. Scene-Single-Direct demands that agents infer tasks from goal images and plan accordingly, while drawers and windows introduce geometric constraints requiring exact manipulation. Scene-Single-Composite elevates this complexity by enforcing causal dependencies, such as a prerequisite button press; it requires high precision in interacting with diverse mechanisms while managing the compounding errors inherent in sequential subtasks. Finally, Cube-Triple represents the highest spatial complexity, demanding precise sequential planning and highly accurate multi-object manipulation to prevent compounding errors across the three targets. Collectively, these benchmarks rigorously evaluate the frameworks in sequential, multi-stage, and vision-guided control settings.

**Training Settings.** Following Maes et al. (2026a), we collect a 2-million-frame "play" dataset for each environment configuration (Cube-Single, Cube-Triple, and Scene-Single). Note that we use the model trained on Scene-Single dataset to evaluate both of Single-Single-Direct and Scene-Single-Composite. These datasets feature unannotated, randomized interactions with no explicit rewards or goal annotations. We train separate Object-Centric Encoders, Dynamics Models, Diffusion Policies, and Contact Predictors per environment on $224 \times 224$ images. During dynamics model training, each training instance pairs a frame with 4 context frames randomly sampled from the preceding 100-frame window. We train for 5 epochs with a batch size of 128 ($\approx 15,500$ steps per epoch) on a single NVIDIA H100 GPU node. Complete training schedules and hyperparameters are in the Appendix.

**Baselines.** We evaluate WorldDP against a diverse set of baselines, with details provided in the Appendix. These include vision-based JEPA world models, specifically **DinoWM** (Zhou et al., 2024) and **LeWM** (Maes

Table 1: Success rate (%) on Cube-Triple and Scene-Single-Composite. 1-Cube, 2-Cubes, and 3-Cubes in Cube-Triple refer to at least 1, 2, and all 3 cube successes. Mean and std reported across 10 seeds.

| Methods | Cube-Triple | | | Scene-Single-Composite | |
|---|---|---|---|---|---|
| | 1-Cube | 2-Cubes | 3-Cubes | Button Press | Full Task |
| DinoWM (Zhou et al., 2024) | $60.4 \pm 4.6$ | $11.8 \pm 4.3$ | $0.8 \pm 1.0$ | $3.4 \pm 2.1$ | $0.2 \pm 0.6$ |
| LeWM (Maes et al., 2026b) | $67.0 \pm 5.1$ | $10.4 \pm 2.1$ | $0.0 \pm 0.0$ | $4.0 \pm 2.5$ | $0.2 \pm 0.6$ |
| HWM (Zhang et al., 2026) | $56.4 \pm 5.4$ | $7.0 \pm 2.9$ | $0.6 \pm 1.0$ | $1.8 \pm 2.0$ | $0.6 \pm 1.3$ |
| HWM[†] (Zhang et al., 2026) | $61.2 \pm 5.1$ | $9.4 \pm 1.9$ | $0.4 \pm 0.8$ | $24.2 \pm 6.8$ | $8.2 \pm 3.9$ |
| HECRL[*] (Haramati et al., 2026) | $75.2 \pm 5.6$ | $33.0 \pm 7.7$ | $4.8 \pm 3.7$ | $55.4 \pm 4.8$ | $19.0 \pm 4.1$ |
| DP40 (Chi et al., 2025) | $65.4 \pm 1.3$ | $9.4 \pm 1.0$ | $0.0 \pm 0.0$ | $\mathbf{69.4 \pm 5.2}$ | $0.0 \pm 0.0$ |
| DP100 (Chi et al., 2025) | $76.2 \pm 3.5$ | $30.4 \pm 5.1$ | $2.2 \pm 1.1$ | $63.8 \pm 6.3$ | $9.2 \pm 2.9$ |
| **WorldDP (Ours)** | $\mathbf{92.8 \pm 2.5}$ | $\mathbf{70.6 \pm 5.0}$ | $\mathbf{29.4 \pm 4.6}$ | $51.0 \pm 10.2$ | $\mathbf{20.2 \pm 5.4}$ |

Table 2: Success rate (%) on Cube-Single and Scene-Single-Direct. Mean and std reported across 10 seeds.

| Methods | Cube | Scene-Single-Direct | | | | Both Task |
|---|---|---|---|---|---|---|
| | Single | Button | Drawer | Window | Average | Average |
| DinoWM (Zhou et al., 2024) | $0.0 \pm 0.0$ | $16.2 \pm 9.4$ | $25.9 \pm 9.3$ | $19.4 \pm 10.0$ | $20.5 \pm 4.3$ | $10.3 \pm 2.2$ |
| LeWM Maes et al. (2026b) | $0.4 \pm 1.3$ | $16.9 \pm 4.2$ | $4.1 \pm 5.6$ | $19.4 \pm 8.3$ | $13.5 \pm 2.8$ | $6.9 \pm 1.7$ |
| HWM Zhang et al. (2026) | $1.0 \pm 1.4$ | $0.6 \pm 2.0$ | $8.8 \pm 6.9$ | $3.5 \pm 4.1$ | $4.3 \pm 2.6$ | $2.7 \pm 1.5$ |
| HWM[†] Zhang et al. (2026) | $0.2 \pm 0.6$ | $43.1 \pm 15.2$ | $48.8 \pm 8.3$ | $21.2 \pm 6.9$ | $37.7 \pm 6.7$ | $19.0 \pm 3.3$ |
| HECRL[*] (Haramati et al., 2026) | $\mathbf{99.0 \pm 1.4}$ | $50.0 \pm 8.8$ | $48.8 \pm 10.8$ | $8.8 \pm 3.1$ | $35.9 \pm 4.3$ | $67.4 \pm 2.2$ |
| DP40 (Chi et al., 2025) | $0.0 \pm 0.0$ | $\mathbf{81.2 \pm 6.6}$ | $27.6 \pm 4.0$ | $6.5 \pm 3.3$ | $38.5 \pm 1.9$ | $19.2 \pm 0.9$ |
| DP100 (Chi et al., 2025) | $98.0 \pm 0.9$ | $66.9 \pm 7.8$ | $21.2 \pm 5.0$ | $4.7 \pm 3.7$ | $30.9 \pm 2.8$ | $64.5 \pm 1.3$ |
| **WorldDP (Ours)** | $71.2 \pm 6.6$ | $65.6 \pm 13.6$ | $\mathbf{75.3 \pm 8.2}$ | $\mathbf{63.5 \pm 11.4}$ | $\mathbf{68.1 \pm 7.3}$ | $\mathbf{69.7 \pm 4.0}$ |

et al., 2026b), alongside hierarchical world models such as **HWM** (Zhang et al., 2026) and an adapted variant, **HWM**[†], which pairs our high-level world model with HWM's 1-step low-level action planner. Additionally, we evaluate **HECRL**[*], a hierarchical framework adapted from Haramati et al. (2026) that integrates our low-level diffusion policy. Finally, we compare against two **Goal-Conditioned Diffusion Policy** baselines (Chi et al., 2025) trained to execute 40-step (**DP40**) and 100-step (**DP100**) trajectories. All baselines are trained on the same trajectories as WorldDP, utilizing the hyperparameter configurations specified in their respective original works wherever applicable.

**Evaluation Settings.** We evaluate all frameworks on distinct, held-out test datasets across 50 trajectories per task over 10 different seeds; the same set of test trajectories is evaluated across all baselines to ensure fairness. Task success is evaluated via ground-truth environment states. For Cube-Single and Cube-Triple, success is determined by the Euclidean distance between the current and target coordinates of the cube(s). For Scene-Single-Direct and Scene-Single-Composite, success is measured by the states of the drawer, sliding window, and push buttons. Comprehensive criteria for success detection are in the Appendix.

## 4.2 Quantitative Results

**Multi-Stage Manipulation Tasks.** The evaluations on the harder tasks (Cube-Triple and Scene-Single-Composite) are presented in Table 1. For Cube-Triple, we decompose performance into the success rates of placing at least one, at least two, and all three cubes. For Scene-Single-Composite, we evaluate the initial button-pressing success and the overall task completion. WorldDP consistently outperforms all baselines, with it achieving significantly higher success at 3 cubes manipulation in the Cube-Triple environment.

We observe a similar trend in the simpler Cube-Single and Scene-Single-Direct environments (Table 2). Because each Scene-Single-Direct episode modifies exactly one environmental entity (a button, the drawer, or the window), we break down its success rates by these respective interaction categories. WorldDP achieves

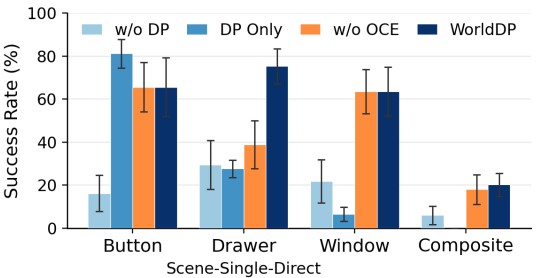

Figure 5: **Ablation Studies:** Removing either the low-level Diffusion Policy (w/o DP) or the Object-Centric Encoder (w/o OCE) degrades success rates compared to our full WorldDP framework.

Table 3: **Ablation study** showing that optimizing during task execution using image embedding cost function degrades performance compared to WorldDP's optimization w.r.t. object embeddings.

| Optim. | Scene-Single | Cube-Triple | | |
|---|---|---|---|---|
| Methods | Composite | 1-Cube | 2-Cubes | 3-Cubes |
| Image Emb. | 17±6 | 74±4 | 26±5 | 3±2 |
| **WorldDP** | **20±5** | **93±3** | **71±5** | **29±5** |

the highest overall performance on these tasks. DP100 and HECRL* (which uses DP100 to track subgoals) excels on Cube-Single due to the task's simplicity. Nevertheless, we outperform all baselines on Scene-Single-Direct on average. LeWM and DinoWM, lacking multi-stage execution capabilities, struggle across all tasks. Their marginal success on Cube-Triple's 1-cube and 2-cube metrics is primarily because some cubes initialize near their target goals and do not need to be moved. WorldDP also outperforms HWM and HWM$^\dagger$ on all the tasks. Notably, HWM$^\dagger$, which includes WorldDP 's high-level model, performs better than HWM.

**Unifying World Model with Diffusion Policy.** To isolate the impact of integrating world model planning with our diffusion policy, we compared our framework against two baselines: "w/o DP," which directly optimizes raw environment actions, and "DP Only," the standalone Diffusion Policy (DP40). Our complete hierarchical framework outperforms both ablated configurations across nearly all tasks, particularly those requiring even basic multi-stage planning like drawer and window manipulation. The method using DP only performs marginally better only on the simple button-pressing task. Ideal subgoals for WorldDP align with keyframes, but predicted targets are often slightly imprecise. While "w/o DP" method fails under these suboptimal subgoals, our low-level diffusion policy robustly executes the transitions to reach them.

Additionally, we evaluate the effect of the policy horizon on our method by replacing our default 40-step diffusion policy with a 100-step variant (DP100). This configuration yields slightly lower performance, achieving an average success rate of *58%* on Scene-Single-Direct and *8%* on Scene-Single-Composite, compared to *74%* and *20%* with our proposed method, respectively. Because our upper-tier world model generates closely spaced, precise subgoals, the shorter 40-step policy horizon provides more effective local action execution.

**Impact of Object-Centric Encoding.** To study the impact of object-centric encoding, we evaluate a variant trained on Scene-Single using raw DINOv2 patch-level embeddings as states following Zhou et al. (2024) ("w/o OCE" in Fig. 5). Our full framework shows more consistent performance compared to this baseline. Additionally, Table 3 compares our method against a variant that optimizes using image embeddings rather than object embeddings. Optimizing via image embeddings degrades performance on both the tasks of Scene-Single-Composite and Cube-Triple.

**Which components are crucial for WorldDP's success?** While the hierarchical unification of world-model planning with DP's low-level execution and object-centric states form the backbone of WorldDP, we also evaluate the individual contributions of the PF and the Contact Predictor (CP) by testing two variants: "w/o PF" (using CEM instead) and "w/o CP." Both modifications degrade performance (Fig. 6), with the CP's omission causing large success rate drops, underscoring its vital role.

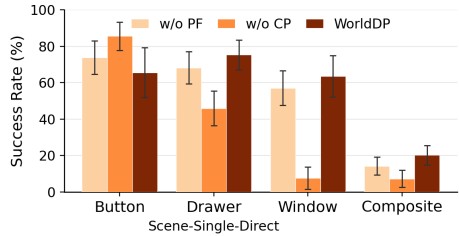

Figure 6: Ablations on use of Particle Filter (PF) and Contact Predictor (CP).

**Generalization to Unseen Environment Conditions.** We evaluate zero-shot generalization across four unseen variations (visual examples in Appendix): (a) *camera viewpoints* (random yaw/pitch within $[-5°, 5°]$), (b) *floor color*, (c) *lighting variations* (intensity in $[0, 1]$), and (d) *object color*. We compare WorldDP against the top baselines (HECRL*, DP40, DP100) across 10 sequences per task over 3 seeds (Table 4). Across Cube-Triple,

Table 4: Generalization to unseen environment conditions: **Camera** Viewpoint Perturbations; Random **Floor** Color; Random **Lighting**; and Random **Object** Color.

| Task Setting | Method | Default | Camera | Floor | Lighting | Object |
|---|---|---|---|---|---|---|
| **Cube-Triple** | DP40 | $0.0 \pm 0.0$ | $0.0 \pm 0.0$ | $0.0 \pm 0.0$ | $0.0 \pm 0.0$ | $0.0 \pm 0.0$ |
| | DP100 | $0.0 \pm 0.0$ | $3.3 \pm 5.8$ | $0.0 \pm 0.0$ | $0.0 \pm 0.0$ | $0.0 \pm 0.0$ |
| | HECRL* | $0.0 \pm 0.0$ | $10.0 \pm 10.0$ | $3.3 \pm 5.8$ | $13.3 \pm 15.3$ | $\mathbf{3.3 \pm 5.8}$ |
| | **WorldDP (Ours)** | $\mathbf{20.0 \pm 17.3}$ | $\mathbf{13.3 \pm 15.3}$ | $\mathbf{6.7 \pm 5.8}$ | $\mathbf{30.0 \pm 10.0}$ | $0.0 \pm 0.0$ |
| **Scene-Single-Composite** | DP40 | $0.0 \pm 0.0$ | $0.0 \pm 0.0$ | $0.0 \pm 0.0$ | $0.0 \pm 0.0$ | $0.0 \pm 0.0$ |
| | DP100 | $10.0 \pm 17.3$ | $0.0 \pm 0.0$ | $\mathbf{3.3 \pm 5.8}$ | $6.7 \pm 11.6$ | $3.3 \pm 5.8$ |
| | HECRL* | $\mathbf{40.0 \pm 0.0}$ | $\mathbf{3.3 \pm 5.8}$ | $0.0 \pm 0.0$ | $10.0 \pm 10.0$ | $10.0 \pm 10.0$ |
| | **WorldDP (Ours)** | $10.0 \pm 10.0$ | $\mathbf{3.3 \pm 5.8}$ | $0.0 \pm 0.0$ | $\mathbf{16.7 \pm 20.8}$ | $\mathbf{20.0 \pm 10.0}$ |
| **Cube-Single** | DP40 | $0.0 \pm 0.0$ | $0.0 \pm 0.0$ | $0.0 \pm 0.0$ | $0.0 \pm 0.0$ | $0.0 \pm 0.0$ |
| | DP100 | $\mathbf{100.0 \pm 0.0}$ | $\mathbf{80.0 \pm 10.0}$ | $\mathbf{83.3 \pm 5.8}$ | $\mathbf{100.0 \pm 0.0}$ | $\mathbf{100.0 \pm 0.0}$ |
| | HECRL* | $\mathbf{100.0 \pm 0.0}$ | $\mathbf{80.0 \pm 10.0}$ | $23.3 \pm 15.3$ | $93.3 \pm 5.8$ | $53.3 \pm 25.2$ |
| | **WorldDP (Ours)** | $83.3 \pm 5.8$ | $66.7 \pm 15.3$ | $46.7 \pm 11.6$ | $76.7 \pm 23.1$ | $76.7 \pm 5.8$ |
| **Scene-Single-Direct** | DP40 | $40.0 \pm 10.0$ | $13.3 \pm 5.8$ | $3.3 \pm 5.8$ | $40.0 \pm 10.0$ | $13.3 \pm 5.8$ |
| | DP100 | $36.7 \pm 5.8$ | $20.0 \pm 17.3$ | $0.0 \pm 0.0$ | $36.7 \pm 5.8$ | $10.0 \pm 0.0$ |
| | HECRL* | $40.0 \pm 0.0$ | $16.7 \pm 20.8$ | $10.0 \pm 10.0$ | $23.3 \pm 5.8$ | $20.0 \pm 17.3$ |
| | **WorldDP (Ours)** | $\mathbf{66.7 \pm 15.3}$ | $\mathbf{50.0 \pm 10.0}$ | $\mathbf{40.0 \pm 17.3}$ | $\mathbf{66.7 \pm 20.8}$ | $\mathbf{63.3 \pm 5.8}$ |

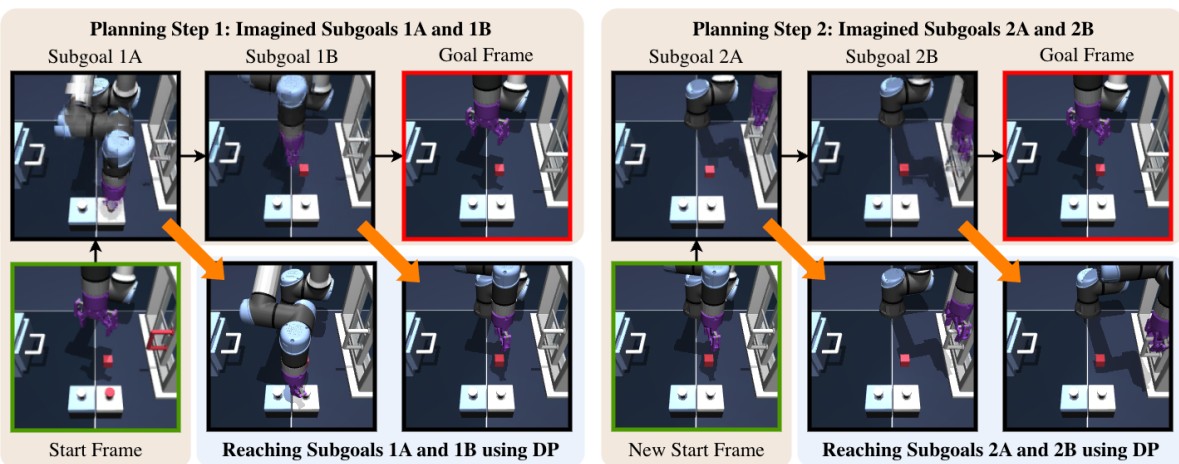

Figure 7: **Task Execution Example.** Given the initial and target states, our framework decomposes the task into sequential planning phases. Step 1 optimizes subgoals (1A, 1B) for the buttons which are executed by the Diffusion Policy (DP) (represented by orange arrows). Step 2 takes the new obvervation as input and optimizes subgoals (2A, 2B) for the drawer and window, subsequently executed by the DP.

Scene-Single-Composite, and Scene-Single-Direct, WorldDP outperforms all baselines under almost every condition, while maintaining higher relative robustness on Cube-Single compared to baselines like HECRL*.

### 4.3 Qualitative Visualizations

**Task Execution.** Fig. 7 shows an example on Scene-Single-Composite. Given initial and goal frames, the planner first optimizes latent actions relative to button embeddings (step 1), producing subgoals 1A and 1B for the low-level DP to execute. Taking the resulting state, step 2 optimizes for drawer and window embeddings to generate subgoals 2A and 2B. This hierarchical structure decomposes complex, multi-stage tasks into manageable sub-problems for the low-level policy. Additional figures are in the Appendix.

**OCE predictions.** Examples of the predicted slot masks $\hat{m}_k$ are shown in Fig. 8. As our OCE operates on DINOv2 patch features, masks are constrained to patch-level rather than pixel-level accuracy, naturally resulting in coarser, block-like boundaries around objects. Additional visualizations are in the Appendix.

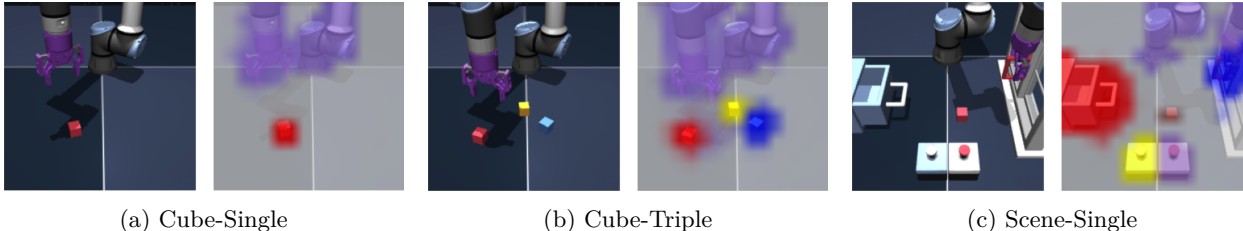

(a) Cube-Single        (b) Cube-Triple        (c) Scene-Single

Figure 8: **Object-Centric Encoding Visualization.** Left: image; Right: pred. masks from OCE emb.

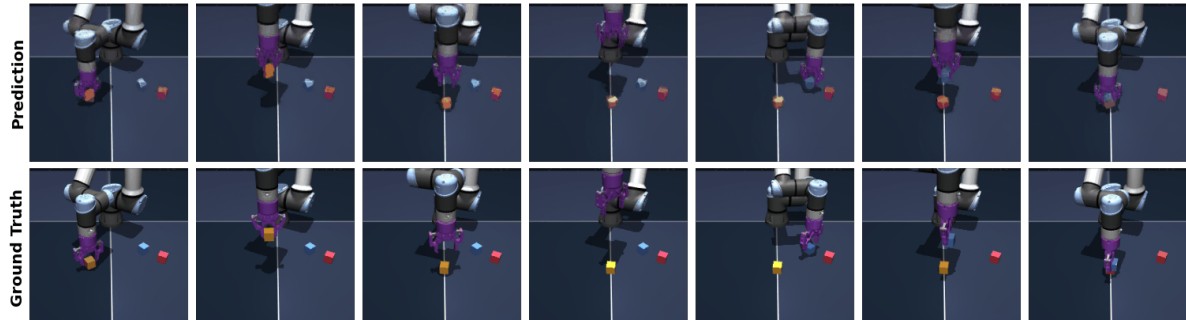

Figure 9: **Open-Loop Trajectory Rollouts.** Given the initial state and an action sequence, we show predicted future states over a 4-second horizon. Latent states are decoded into images for visualization. Predicted frames are subsampled in the figure due to space constraints.

**World Model Rollouts.** An open-loop rollout on Cube-Triple is illustrated in Fig. 9. Given an initial frame and ground-truth action sequence, our world model generates predicted trajectories that closely match ground-truth execution, confirming its accuracy. Note that while planning occurs entirely in latent object-embedding space, a separate Transformer-based decoder (used solely for visualization) reconstructs latents into images. Additional visualizations are in the Appendix.

## 5 Research Impact and Limitations

We present WorldDP, a framework combining a high-level world model with a low-level diffusion policy to improve multi-stage task planning and execution. Further, the use of particle filters is new for MPC-based planning using learned world models and naturally captures the multi-modal nature of robotic tasks. Despite its success, some limitations remain. Relying on image goals is restrictive, making language goals a promising future direction. Test-time particle filter sampling also introduces up to two minutes of initial overhead, which gradient-based or parallelized optimization could mitigate. Further, replacing our random training-frame sampling with smart sampling strategies could boost performance. While WorldDP assumes a fixed object count, it can adapt to varying counts by setting a sufficiently large slot limit. During planning, the model can then focus exclusively on the slots exhibiting the most changes between the start and goal frames. Finally, extending the model to deformable or splitting entities also remains future work directions.

## 6 Conclusion

In this work, we introduced WorldDP, a hierarchical world model framework tailored for multi-stage robotic tasks. WorldDP leverages an object-centric world model as a high-level planner to decompose complex sequential tasks into simpler subgoals, which are subsequently executed by a low-level diffusion policy. To optimize these subgoals during planning, we integrated a particle filter alongside a contact prediction loss. Extensive evaluations across four diverse task settings demonstrate that WorldDP consistently outperforms existing baselines. Ultimately, our findings underscore the power of unifying long-term world model planning with robust, short-term diffusion policies to achieve superior generalization. We hope these insights inspire further research at the intersection of world models and robotics, with possible extensions of our work being in scaling model capacity and cross-embodiment learning.

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

## A  Model Hyperparameter

### A.1  Object-Centric Encoder

The object-centric encoder parameters for each of the datasets is shown in Table 5. For Cube-Single, the three slots correspond to the robot, the background, and the cube. In Cube-Triple, the five slots represent the robot, the background, and the three cubes, while for Scene-Single, the seven slots account for the robot, background, drawer, two buttons, window, and cube. Models for all datasets are trained for 10 epochs with a batch size of 64. Optimization is performed using the AdamW optimizer (Loshchilov & Hutter, 2017) with an initial learning rate of $10^{-3}$, which is decayed to $10^{-6}$ via a one-cycle learning rate scheduler with cosine annealing (Smith & Topin, 2019).

Table 5: Hyperparameter configurations for the Object-Centric Encoder.

| Dataset | Num Slots | Slot Dim. | Slot Corrector | | Slot Decoder | |
|---|---|---|---|---|---|---|
| | | | Num. iter. | Hidden Dim. | Layers | Hidden Dim. |
| Cube-Single | 3 | 64 | 3 | 128 | 3 | 384 |
| Cube-Triple | 5 | 128 | 3 | 128 | 3 | 384 |
| Scene-Single | 7 | 1024 | 3 | 128 | 3 | 384 |

### A.2  Conditional Diffusion Transformer Dynamics Prediction Model.

As detailed in Section 3.2, the CDiT takes object-centric states as input; thus, its input dimension matches the slot dimension of each respective dataset. All other CDiT parameters are held constant across datasets, featuring a depth of 12, 4 attention heads, and an MLP ratio of 2.0. The transformer-based action encoder maps the 5-dimensional physical action space to a 32-dimensional latent action space, utilizing a hidden dimension of 256, a depth of 3, and 4 attention heads. Models for all datasets are trained for 10 epochs with a batch size of 128. Optimization is performed using the AdamW optimizer (Loshchilov & Hutter, 2017) with an initial learning rate of $10^{-4}$, which is decayed to $10^{-7}$ via a one-cycle learning rate scheduler with cosine annealing (Smith & Topin, 2019).

### A.3  Diffusion Policy.

Apart from the input dimension, which scales with the slot dimension and the number of slots, all other parameters of the diffusion policy transformer remain constant across datasets. Specifically, we use a prediction horizon of 40, a depth of 8, 4 attention heads, and a hidden dimension of 256. For all datasets, models are trained for 3 epochs with a batch size of 48. Optimization is performed using the AdamW optimizer (Loshchilov & Hutter, 2017) with an initial learning rate of $10^{-4}$ and a weight decay of $10^{-6}$.

### A.4  Contact Predictor

The contact predictor consists of two linear layers separated by a SiLU activation function (Elfwing et al., 2018). It takes the mean of the DINOv2 patch-level representations as input and outputs a vector corresponding to the number of manipulable objects in the environment: 1 for Cube-Single, 3 for Cube-Triple, and 5 for Scene-Single. Across all datasets, the model is trained for 10 epochs with a batch size of 256. Optimization is performed using the AdamW optimizer (Loshchilov & Hutter, 2017) with an initial learning rate of $10^{-3}$, which decays to $10^{-6}$ via a one-cycle learning rate scheduler with cosine annealing (Smith & Topin, 2019).

## B  Tversky Loss

As described in Section 3.1, to supervise predicted masks during object-centric encoder training, we employ the Tversky loss function (Salehi et al., 2017). Given predicted $(\hat{m}_k)$ and ground-truth $(m_k)$ masks, the

Tversky loss is formulated as:

$$\mathcal{L}_{\text{tv}} = 1 - \frac{TP + \epsilon}{TP + \alpha FN + \beta FP + \epsilon} \tag{5}$$

where $TP = \sum \hat{m}_k m_k$ represents true positives, $FP = \sum \hat{m}_k(1-m_k)$ false positives, and $FN = \sum(1-\hat{m}_k)m_k$ false negatives. The hyperparameters $\alpha = 0.99$ and $\beta = 0.01$ are used to control the trade-off between false positives and false negatives, with $\epsilon = 10^{-6}$ for numerical stability. To further prioritize small objects, we apply an inverse-size weighting scheme, scaling each object's loss contribution by $(1/\text{size})^{\text{temperature}}$ (with temperature $= 0.2$) and normalizing these weights across the batch.

## C   Planning Parameters

The hyperparameter choices utilized for the particle-filter-based trajectory optimization across all datasets are detailed in Table 6, where $T$ denotes the planning horizon, $Q$ is the total number of particles, $\sigma$ represents the action-sampling standard deviation, $M$ is the elite size (number of top candidates selected for resampling), $L$ signifies the number of optimization iterations, and $\lambda_{plan}$ balances the cost objectives. Moreover, as mentioned in the main text, for Cube-Triple, we run and execute the optimization with respect to each cube sequentially; for Scene-Single-Composite, we first optimize with respect to the button entities and subsequently with respect to the drawer and window components.

Table 6: Optimization hyperparameters for the Particle Filter planner across different datasets.

| Dataset | $T$ | $Q$ | $\sigma$ | $M$ | $L$ | $\lambda_{plan}$ |
|---|---|---|---|---|---|---|
| Cube-Single | 3 | 600 | 1.0 | 10 | 20 | 0.1 |
| Cube-Triple | 3 | 600 | 0.5 | 10 | 10 | 0.05 |
| Scene-Single-Direct | 2 | 600 | 0.1 | 10 | 10 | 0.1 |
| Scene-Single-Composite | 2 | 1200 | 0.1 | 10 | 10 | 1 |

**Planning/Execution Time.** The execution times (including planning for the world-model-based method) per episode are reported in Table 7. These are reported for execution on a single A4000 GPU with 16 GB of GPU memory. While WorldDP 's planning phase introduces higher test-time overhead (65 seconds) than baselines like DP and HECRL, it yields higher success rates on multi-stage tasks. Crucially, once subgoals are planned, the policy executes actions in real-time without latency. Moreover, the planning time can be further reduced through parallel computing over several GPUs. Furthermore, our particle-filter optimization and CEM receives the same rollout and planning budget in our experiments and incurs roughly the same computational runtime.

Table 7: Planning/Exectution Times

| Method | Time (s) |
|---|---|
| DinoWM (Zhou et al., 2024) | 237.3 |
| LeWM (Maes et al., 2026b) | 6.9 |
| HWM (Zhang et al., 2026) | 31.1 |
| HWM$^{\dagger}$ (Zhang et al., 2026) | 360 |
| HECRL$^{*}$ (Haramati et al., 2026) | 1.1 |
| WorldDP w/o PF (with CEM) | 64.7 |
| **WorldDP (Ours)** | 64.5 |

## D   Success Rate Calculation

**Cube-Single.** A trial is considered successful if the Euclidean distance between the current cube position and its target position is within 12 cm at the end of the episode.

**Cube-Triple.** This metric extends the single-cube condition to a multi-object setting, requiring the final Euclidean distance for each of the three cubes to be within 12 cm of its respective target position simultaneously.

**Scene-Single-Direct.** Success is achieved if all binary button states match their target configurations, and the final positions of the drawer and window handles are within 4 cm and 3 cm of their respective target coordinates.

**Scene-Single-Composite.** This task shares the identical completion criteria as Scene-Single-Direct, requiring correct binary button states alongside precise positioning of both the drawer and window handles.

## E    Baselines

We detail the baseline methods and their specific implementations in the following.

**DinoWM (Zhou et al., 2024)**: A vision-based JEPA world model framework that utilizes frozen DINOv2 patch features as its state representation. To ensure a fair comparison, all core model parameters are preserved while modifying the training data sequence to a horizon of 100 frames (matching WorldDP) with action chunks of size 25. Planning utilizes a receding-horizon approach with a 2-step execution window per optimization. The total planning budget varies by task complexity: 4 steps for Cube-Single and Scene-Single-Direct (re-planning once after executing the first 2 steps), 12 steps for Cube-Triple, and 8 steps for Scene-Single-Composite. Following the Cube-Single configuration in Maes et al. (2026b), the CEM optimizer runs for 10 iterations per step, sampling 300 candidates and selecting the top 30 elites.

**LeWM (Maes et al., 2026b)**: A JEPA world model that jointly optimizes its image encoder and latent dynamics model. Its highly compact state space enables accelerated planning times. Model hyperparameters match the original Cube-Single implementation. Training horizons, action chunking, and receding-horizon planning setups are identical to those described for DinoWM above.

**HWM** (Zhang et al., 2026): A hierarchical JEPA world model that plans by first optimizing for high-level intermediate subgoals and then determining low-level actions to reach them. Hyperparameters are kept as close as possible to the official open-source implementation.

**HWM**[†] (Zhang et al., 2026): This is an adapted variant modified for our benchmark, as the original HWM release targets a smaller environment. HWM[†] pairs WorldDP's high-level world model, weights, and planning framework with HWM's 1-step low-level transition model to explicitly plan for action sequences toward subgoals. During execution, its low-level planner optimizes over a 25-step horizon to match the action prediction window of WorldDP's low-level diffusion policy.

**HECRL**[*]: An adaptation of the HECRL framework (Haramati et al., 2026). The original architecture uses a diffusion model to predict subgoals, which are tracked by a goal-conditioned RL agent. For a fair comparison, our modified HECRL[*] replaces the low-level RL controller with our goal-conditioned diffusion policy, uses single-view image inputs instead of multi-view streams, and receives the same budget of environmental feedback during inference as our method. The core idea behind HECRL is to generate subgoals using diffusion, given start and goal frames, with the subgoals being reached by a goal-conditioned low-level RL controller. Therefore, in HECRL[*], for a direct comparison, we use the same HECRL framework for subgoal generation but replace their goal-conditioned RL controller with our goal-conditioned diffusion policy which takes the same inputs and outputs low-level actions.

**Goal Conditioned Diffusion Policy (DP) (Chi et al., 2025)**: A standard imitation learning baseline for robotic manipulation, structurally identical to the low-level policy tracker in WorldDP. We evaluate two horizons: **DP40**, trained to execute trajectories of 40 steps, and **DP100**, optimized for trajectories of 100 steps. All implementation and architectural details are as described in the main text and the diffusion policy overview in Section A.3.

As mentioned in Section 3.4, the end-effector depth-camera pose correction is integrated directly into the low-level Diffusion Policy pipeline for the Cube-Single and Cube-Triple tasks. Consequently, the Diffusion Policy baselines (DP40, DP100) and HECRL[*] (which uses Diffusion Policy in its lower hierarchy) receive

Table 8: Comparison of supervision and information required by each method during training and evaluation.

| Method | Training | | | Evaluation |
| --- | --- | --- | --- | --- |
| | Expert Trajectories | Segmentation Masks (SAM2) | Contact Labels | Depth-Camera Pose Correction |
| DinoWM (Zhou et al., 2024) | ✓ | | | |
| LeWM (Maes et al., 2026b) | ✓ | | | |
| HWM (Zhang et al., 2026) | ✓ | | | |
| HWM$^\dagger$ (Zhang et al., 2026) | ✓ | ✓ | ✓ | |
| HECRL* (Haramati et al., 2026) | ✓ | ✓ | | ✓ (Cube tasks only) |
| DP40 / DP100 (Chi et al., 2025) | ✓ | ✓ | | ✓ (Cube tasks only) |
| **WorldDP (Ours)** | ✓ | ✓ | ✓ | ✓ (Cube tasks only) |

this exact same correction. The DinoWM, LeWM, and HWM baselines do not use Diffusion Policy in any capacity and thus do not receive it. Crucially, WorldDP still outperforms all baselines in the Scene-Single-Direct and Scene-Single-Composite tasks on average, where the depth-camera correction is not used at all. Table 8, further, lists all the supervision used by each method. Notably, because the Diffusion Policy use the same object-centric state representation as WorldDP, baselines HECRL*, DP40, and DP100 also require segmentation masks to train their state encoder.

## F  Additional Visualizations

**Task Execution.**  Along with the visualization of Scene-Single-Composite in Section 4.3, we show task execution visualizations for Cube-Single, Scene-Single-Direct, and Cube-Triple in Figure 10.

**OCE Predictions.**  Additional visualizations for predicted masks for each of the datasets are shown in Figures 11, 12, and 13.

**World Model Rollouts.**  Additional rollout visualizations for each dataset are shown in Figure 14.

**Generalization Experiments Examples.**  We show examples (on Scene-Single) of each of the variations for the generalization experiments of Section 4.2 in Figure 15.

**Impact of mask-based loss during Object-Centric-Encoder learning.**  Figure 16 shows an example of the difference without and with the loss $\mathcal{L}_{tv}$ to supervise the predicted masks.

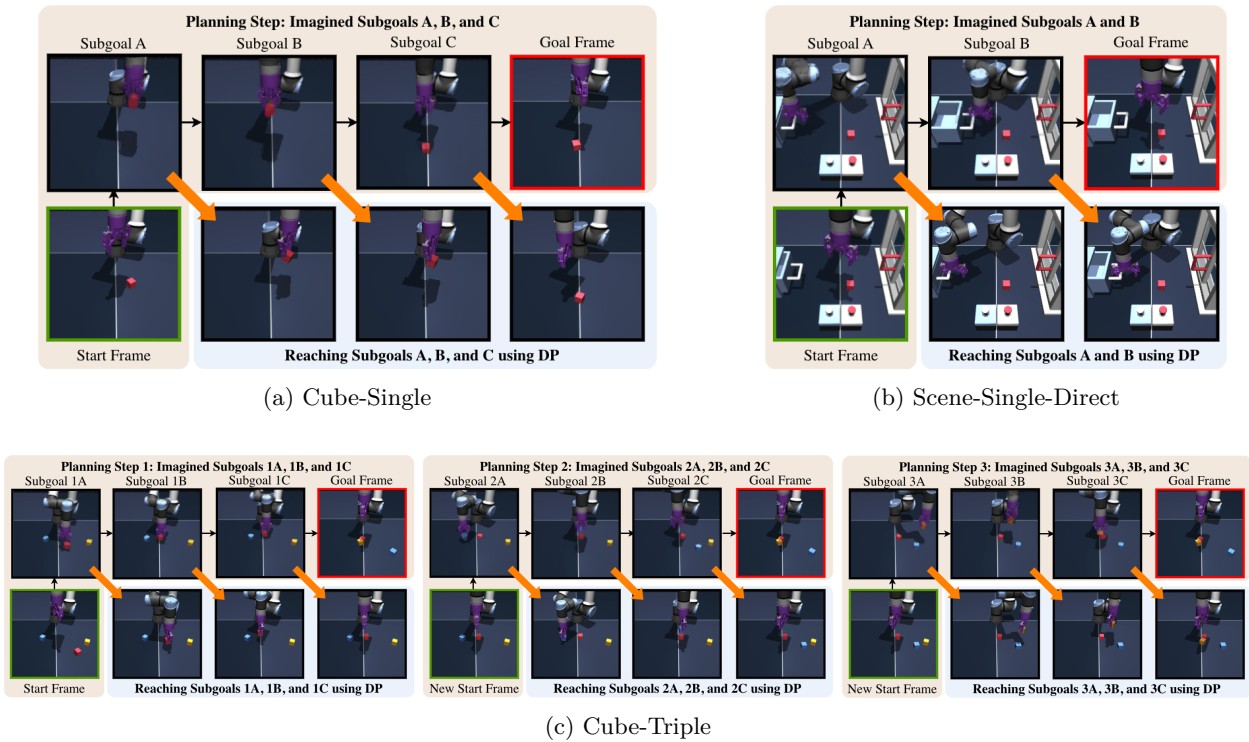

(a) Cube-Single

(b) Scene-Single-Direct

(c) Cube-Triple

Figure 10: **Task Execution Examples.** Given the initial and target states, our framework decomposes the task into sequential planning phases, with each finding a set of subgoals, which are reached using the diffusion policy.

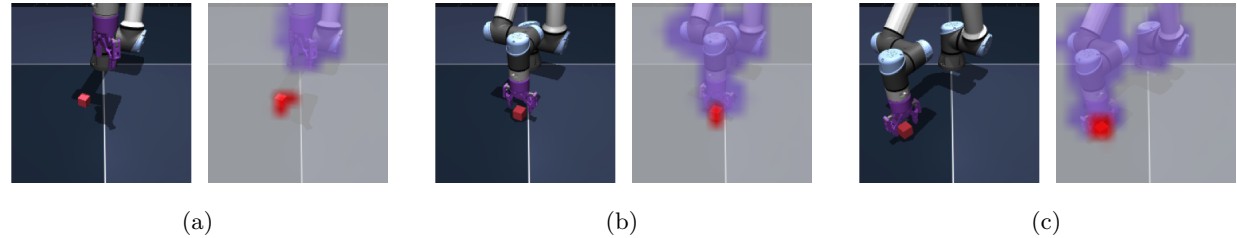

(a)        (b)        (c)

Figure 11: **Object-Centric Encoding Visualization for Cube-Single.** Left: image; Right: predicted masks from OCE embeddings

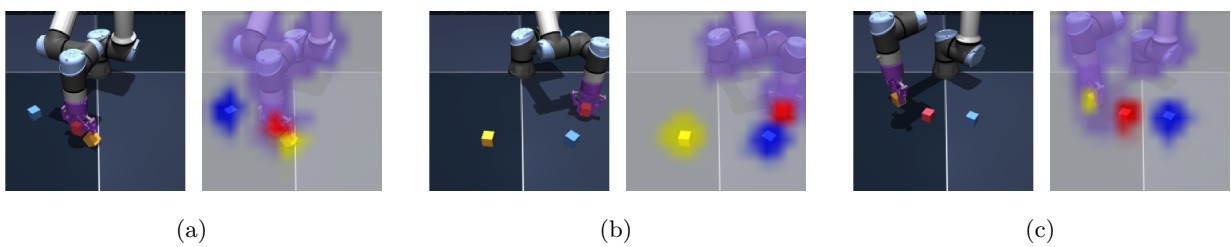

(a)        (b)        (c)

Figure 12: **Object-Centric Encoding Visualization for Cube-Triple.** Left: image; Right: predicted masks from OCE embeddings

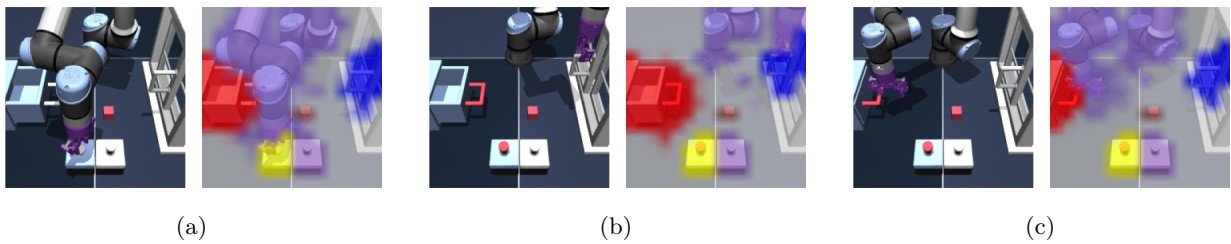

(a)           (b)           (c)

Figure 13: **Object-Centric Encoding Visualization for Scene-Single.** Left: image; Right: predicted masks from OCE embeddings

(a) Cube-Single

(b) Cube-Triple

(c) Scene-Single

Figure 14: **Open-Loop Trajectory Rollouts.** Given the initial state and an action sequence, we show future states over a 4-second horizon. Latent states are decoded into images for visualization. Predicted frames are subsampled in the figure due to space constraints.

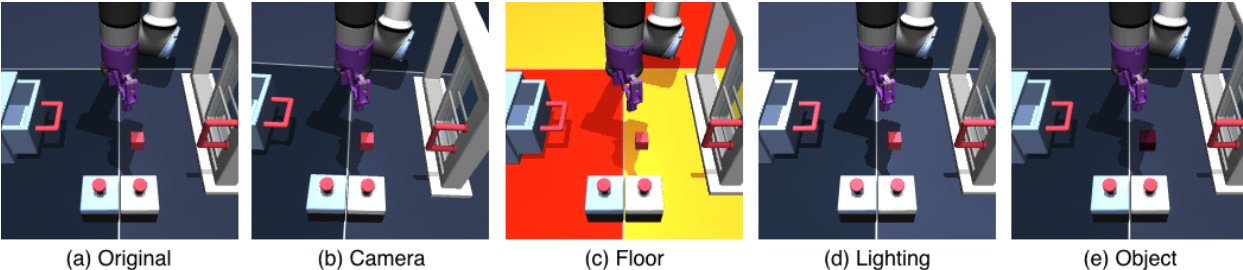

Figure 15: Examples (on Scene-Single) of each of the variations for the generalization experiments of Section 4.2: **Camera** Viewpoint Perturbations; Random **Floor** Color; Random **Lighting**; and Random **Object** Color.

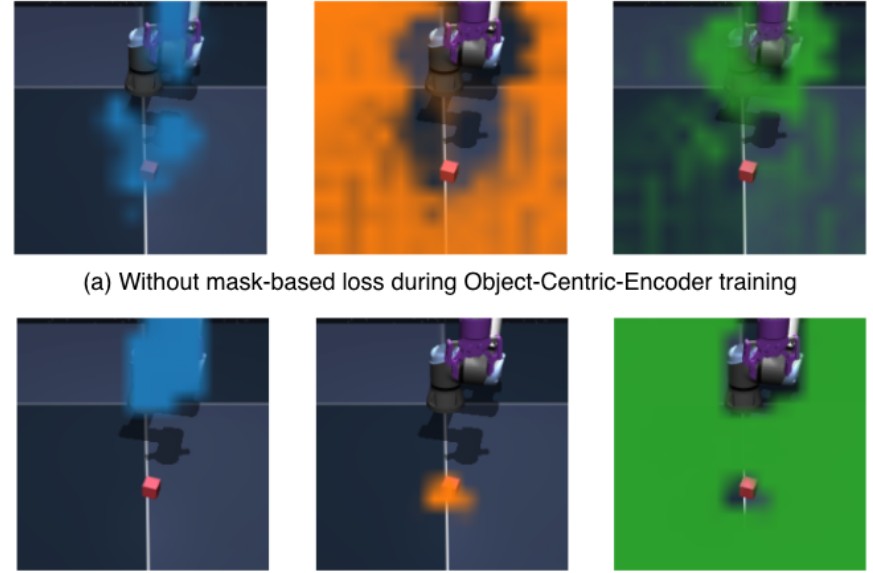

Figure 16: Qualitative example on the impact of the loss $\mathcal{L}_t v$ used to supervise predicted object masks during training of the object-centric encoder. Incorporating $\mathcal{L}_{\text{tv}}$ ensures that predicted slots are more accurately associated with the underlying physical objects, as visually demonstrated by the segmentation masks in this figure.

