# OpenReview forum: "Unifying Object-Centric World Models and Diffusion Policy: A Hierarchical Framework for Multi-Stage Robotic Tasks"
_TMLR — Under review for TMLR_

### Review · Reviewer_xosu · 2026-06-19

**Summary Of Contributions:**

This paper proposes WorldDP, a hierarchical framework for multi-stage robotic manipulation. The method combines an object-centric latent world model for high-level subgoal planning with a goal-conditioned diffusion policy for low-level execution. The object-centric representation is learned on top of frozen DINOv2 patch features using a slot-attention-style encoder with SAM2-generated masks as auxiliary supervision. At test time, the world model is used inside an MPC-style planning loop to optimize latent action sequences and generate subgoals. These subgoals are then sequentially tracked by the diffusion policy. The authors further introduce an object-centric planning cost, a contact-prediction auxiliary cost, and particle-filter-based optimization instead of the more standard CEM optimizer.

**Audience:**

Yes

**Audience Explanation:**

Interesting and timely problem. The paper addresses multi-stage robotic manipulation with visual world models, which is an important limitation of current latent world-model planning approaches. The motivation for combining long-horizon world-model planning with short-horizon diffusion-policy execution is compelling.

Reasonable hierarchical design. The proposed decomposition is natural: the world model handles high-level subgoal generation, while the diffusion policy handles local control. This avoids forcing the world model to optimize low-level continuous actions over long horizons, which is often brittle.

Object-centric planning is well motivated. Using object-centric states is appropriate for manipulation tasks where success depends on changing specific entities. The object-level cost function also makes the planning objective more structured than generic image-level embedding matching.

Empirical gains are meaningful. The reported results show clear improvements on the harder multi-stage tasks, especially Cube-Triple and Scene-Single-Composite. The method appears to improve both over pure world-model planning and over diffusion-policy-only baselines.

Useful ablations. The paper includes ablations for the diffusion-policy component, object-centric encoder, image-embedding versus object-embedding cost, particle filter, contact predictor, and diffusion-policy horizon. These ablations help support the main design choices.

**Claims And Evidence:**

Yes

**Claims Explanation:**

The main claims are generally supported by the experimental evidence. The paper claims that WorldDP improves multi-stage robotic manipulation by combining object-centric world-model planning with a low-level diffusion policy, and the quantitative results support this: WorldDP outperforms world-model baselines, diffusion-policy baselines, and the adapted HECRL* baseline on the harder Cube-Triple and Scene-Single-Composite tasks. The ablation studies also provide useful evidence that the diffusion-policy executor, object-centric representation, particle-filter optimizer, and contact predictor each contribute to the final performance.

However, the evidence is not fully convincing for the broader claims. The experiments are limited to a small number of simulated environments, with relatively structured tasks and fixed visual settings. The paper also does not provide enough statistical information, such as confidence intervals or variance across seeds, which makes it difficult to judge the robustness of some reported gains. In addition, the reliance on SAM2-guided object masks and contact prediction should be clarified further, since these components may involve privileged or external supervision.

Overall, the evidence supports the core empirical claim that WorldDP performs better than the tested baselines in the evaluated benchmark settings. But the broader claims about general multi-stage robotic manipulation and superior generalization should be toned down or supported by additional experiments.

**Requested Changes:**

Weaknesses / Concerns

The novelty is somewhat incremental.
The paper combines several existing ideas: latent world models, object-centric representations, hierarchical subgoal planning, particle-filter/CEM-style sampling-based optimization, and diffusion policies. While the integration is useful and effective, the paper should more clearly distinguish which components are technically novel and which are adapted from prior work.

The evaluation scope is still limited.
The experiments are conducted on a small set of simulated robotic manipulation environments with fixed visual settings and structured tasks. The results are promising, but broader evaluation would strengthen the paper, for example on more diverse objects, randomized visual conditions, unseen task compositions, or real-robot settings.

The dependence on privileged or external supervision needs clarification.
The object-centric encoder is trained with SAM2-generated segmentation masks, and the contact predictor appears to play an important role in planning. The paper should clarify how contact labels are obtained and how robust the method is when segmentation masks or contact annotations are noisy, unavailable, or less accurate.

Statistical reliability and runtime cost need stronger reporting.
The reported success rates are based on a relatively small number of evaluation trajectories, and the paper does not clearly report confidence intervals or variance across seeds. In addition, the particle-filter planning procedure can introduce significant test-time overhead. More detailed reporting of variance, runtime, and compute cost would make the empirical claims more convincing.

---

> ### Author Response · Authors · 2026-07-21
> **Response to Reviewer xosu Part 1/n**
>
> We thank the reviewer for the detailed review and for the constructive comments. We further appreciate your recognition of the strengths of the paper. In the following, we address each of the concerns (weaknesses and requested changes).
>
> **Concern 1: The novelty is somewhat incremental. The paper ... prior work.**
>
> **Response:** WorldDP is the first hierarchical latent-space (JEPA) world model where subgoals are generated in the top hierarchy via particle-filter-based planning on object-centric representations, and reached via a goal-conditioned diffusion policy in the lower hierarchy. While latent-space world models (e.g., Zhou et al., 2024), object-centric representations (e.g., Singh et al., 2025), and Diffusion Policy (Chi et al., 2025) are existing individual paradigms, integrating them into a unified framework for multi-stage tasks requires several novel framework-level contributions:
> - Object-Centric Representation: Existing object-centric encoders that learn over patch-level representations of backbones like DINOv2, often merge small objects with the robot arm into a single entity. We resolve this by adding a mask loss using SAM2-generated masks. Without this, the object-centric representation of our states would be inaccurate.
> - Dynamics Predictor: While our CDiT predictor architecture is similar to Goswami et al. (2025), WorldDP operates directly on object-centric embeddings and includes an action encoder to handle dense actions over long horizons.
> - Particle Filter (PF) Planning: Prior methods (e.g., Zhou et al., 2024) rely on CEM, which uses a single Gaussian distribution and struggles with multi-modal action distributions. We replace CEM with a PF to maintain diverse hypotheses and plan sequentially over object embeddings rather than image patches.
> - Contact Predictor: We use a learned contact predictor to explicitly guide the planning algorithm toward critical manipulation subgoals.
>
> As shown via our ablation studies (where we individually remove these components), a naive integration of these components fails to yield high performance. Instead, these specific technical modifications enable WorldDP to outperform existing world models and diffusion policies on multi-stage tasks (Tables 1 and 2). These distinctions have been further clarified in Section 3 of the manuscript.
>
> **Concern 2: The reported success rates are based on a relatively small number of evaluation trajectories, and the paper does not clearly report confidence intervals or variance across seeds.**
>
> **Response:** We have updated all tables and figures in the revised manuscript to include confidence intervals, reporting the mean and standard deviation across 10 different seeds. The results from Tables 1 and 2 are also shown below. The results follow the original pattern: WorldDP consistently outperforms baseline methods on most tasks while demonstrating the highest consistency, and its strong performance on complex multi-stage tasks (e.g., Cube-Triple) underscores its robustness.
>
> Table 1: Success rate (%) on Cube-Triple and Scene-Single-Composite. 1-Cube, 2-Cubes, and 3-Cubes in Cube-Triple refer to at least 1, 2, and all 3 cube successes. Mean and Standard Deviation are reported over 10 seeds.
> | Method | Cube-Triple (1-Cube) | Cube-Triple (2-Cubes) | Cube-Triple (3-Cubes) | Scene-Single-Composite (Button Press) | Scene-Single-Composite (Full Task) |
> | :--- | :---: | :---: | :---: | :---: | :---: |
> | DinoWM | 60.4±4.6 | 11.8±4.3 | 0.8±1.0 | 3.4±2.1 | 0.2±0.6 |
> | LeWM | 67.0±5.1 | 10.4±2.1 | 0.0±0.0 | 4.0±2.5 | 0.2±0.6 |
> | HWM | 56.4±5.4 | 7.0±2.9 | 0.6±1.0 | 1.8±2.0 | 0.6±1.3 |
> | HWM$^\dagger$ | 61.2±5.1 | 9.4±1.9 | 0.4±0.8 | 24.2±6.8 | 8.2±3.9 |
> | HECRL$^*$ | 75.2±5.6 | 33.0±7.7 | 4.8±3.7 | 55.4±4.8 | 19.0±4.1 |
> | DP40 | 65.4±1.3 | 9.4±1.0 | 0.0±0.0 | **69.4±5.2** | 0.0±0.0 |
> | DP100 | 76.2±3.5 | 30.4±5.1 | 2.2±1.1 | 63.8±6.3 | 9.2±2.9 |
> | **WorldDP (Ours)** | **92.8±2.5** | **70.6±5.0** | **29.4±4.6** | 51.0±10.2 | **20.2±5.4** |
>
>
> Table 2: Success rate (%) on Cube-Single and Scene-Single-Direct tasks. Mean and Standard Deviation are reported over 10 seeds.
> | Method | Cube-Single | Scene-Single-Direct (Button) | Scene-Single-Direct (Drawer) | Scene-Single-Direct (Window) | Scene-Single-Direct (Average) | Both Task Avg |
> | :--- | :---: | :---: | :---: | :---: | :---: | :---: |
> | DinoWM | 0.0±0.0 | 16.2±9.4 | 25.9±9.3 | 19.4±10.0 | 20.5±4.3 | 10.3±2.2 |
> | LeWM | 0.4±1.3 | 16.9±4.2 | 4.1±5.6 | 19.4±8.3 | 13.5±2.8 | 6.9±1.7 |
> | HWM | 1.0±1.4 | 0.6±2.0 | 8.8±6.9 | 3.5±4.1 | 4.3±2.6 | 2.7±1.5 |
> | HWM$^\dagger$ | 0.2±0.6 | 43.1±15.2 | 48.8±8.3 | 21.2±6.9 | 37.7±6.7 | 19.0±3.3 |
> | HECRL$^*$ | **99.0±1.4** | 50.0±8.8 | 48.8±10.8 | 8.8±3.1 | 35.9±4.3 | 67.4±2.2 |
> | DP40 | 0.0±0.0 | **81.2±6.6** | 27.6±4.0 | 6.5±3.3 | 38.5±1.9 | 19.2±0.9 |
> | DP100 | 98.0±0.9 | 66.9±7.8 | 21.2±5.0 | 4.7±3.7 | 30.9±2.8 | 64.5±1.3 |
> | **WorldDP (Ours)** | 71.2±6.6 | 65.6±13.6 | **75.3±8.2** | **63.5±11.4** | **68.1±7.3** | **69.7±4.0** |

---

> > ### Author Response · Authors · 2026-07-21
> > **Response to Reviewer xosu Part 2/n**
> >
> > **Concern 3: The evaluation scope is still limited. The experiments are conducted on a small set of simulated robotic manipulation environments with fixed visual settings and structured tasks. The results are promising, but broader evaluation would strengthen the paper, for example on more diverse objects, randomized visual conditions, unseen task compositions, or real-robot settings.**
> >
> > **Response:** To address this concern, we added generalization experiments in Section 4 of the revised manuscript under four randomized conditions: (a) object colors: the colors of the manipulated objects are randomly changed, (b) camera viewpoints: rotated about its optical center by a random yaw and pitch, each drawn uniformly from [-5$^\circ$, 5$^\circ$] (c) background colors, and (d) lighting variations: the intensity of the global colors are drawn uniformly from [0,1]. Visual examples of each of these are in the Appendix. We compared WorldDP against the top-performing baselines (HECRL*, DP40, DP100) across 10 episodes over 3 seeds for each task. Across Cube-Triple, Scene-Single-Composite, and Scene-Single-Direct, WorldDP outperforms all baselines on almost every condition. On Cube-Single, it retains higher relative robustness under adverse environment shifts than established baselines like HECRL$^*$.
> >
> > Table: Generalization to unseen environment conditions: Camera Viewpoint Perturbations; Random Floor Color; Random Lighting; and Random Object Color.
> > | Task Setting | Method | Default | Camera | Floor | Lighting | Object |
> > | :--- | :--- | :---: | :---: | :---: | :---: | :---: |
> > | **Cube-Triple** | DP40 | 0.0 ± 0.0 | 0.0 ± 0.0 | 0.0 ± 0.0 | 0.0 ± 0.0 | 0.0 ± 0.0 |
> > | | DP100 | 0.0 ± 0.0 | 3.3 ± 5.8 | 0.0 ± 0.0 | 0.0 ± 0.0 | 0.0 ± 0.0 |
> > | | HECRL$^*$ | 0.0 ± 0.0 | 10.0 ± 10.0 | 3.3 ± 5.8 | 13.3 ± 15.3 | **3.3 ± 5.8** |
> > | | **WorldDP (Ours)** | **20.0 ± 17.3** | **13.3 ± 15.3** | **6.7 ± 5.8** | **30.0 ± 10.0** | 0.0 ± 0.0 |
> > | | | | | | | |
> > | **Scene-Single-Composite** | DP40 | 0.0 ± 0.0 | 0.0 ± 0.0 | 0.0 ± 0.0 | 0.0 ± 0.0 | 0.0 ± 0.0 |
> > | | DP100 | 10.0 ± 17.3 | 0.0 ± 0.0 | **3.3 ± 5.8** | 6.7 ± 11.6 | 3.3 ± 5.8 |
> > | | HECRL$^*$ | **40.0 ± 0.0** | **3.3 ± 5.8** | 0.0 ± 0.0 | 10.0 ± 10.0 | 10.0 ± 10.0 |
> > | | **WorldDP (Ours)** | 10.0 ± 10.0 | **3.3 ± 5.8** | 0.0 ± 0.0 | **16.7 ± 20.8** | **20.0 ± 10.0** |
> > | | | | | | | |
> > | **Cube-Single** | DP40 | 0.0 ± 0.0 | 0.0 ± 0.0 | 0.0 ± 0.0 | 0.0 ± 0.0 | 0.0 ± 0.0 |
> > | | DP100 | **100.0 ± 0.0** | **80.0 ± 10.0** | **83.3 ± 5.8** | **100.0 ± 0.0** | **100.0 ± 0.0** |
> > | | HECRL$^*$ | **100.0 ± 0.0** | **80.0 ± 10.0** | 23.3 ± 15.3 | 93.3 ± 5.8 | 53.3 ± 25.2 |
> > | | **WorldDP (Ours)** | 83.3 ± 5.8 | 66.7 ± 15.3 | 46.7 ± 11.6 | 76.7 ± 23.1 | 76.7 ± 5.8 |
> > | | | | | | | |
> > | **Scene-Single-Direct** | DP40 | 40.0 ± 10.0 | 13.3 ± 5.8 | 3.3 ± 5.8 | 40.0 ± 10.0 | 13.3 ± 5.8 |
> > | | DP100 | 36.7 ± 5.8 | 20.0 ± 17.3 | 0.0 ± 0.0 | 36.7 ± 5.8 | 10.0 ± 0.0 |
> > | | HECRL$^*$ | 40.0 ± 0.0 | 16.7 ± 20.8 | 10.0 ± 10.0 | 23.3 ± 5.8 | 20.0 ± 17.3 |
> > | | **WorldDP (Ours)** | **66.7 ± 15.3** | **50.0 ± 10.0** | **40.0 ± 17.3** | **66.7 ± 20.8** | **63.3 ± 5.8** |
> >
> > **Concern 4: The dependence on privileged or external supervision needs clarification. The object-centric encoder is trained with SAM2-generated segmentation masks, and the contact predictor appears to play an important role in planning. The paper should clarify how contact labels are obtained and how robust the method is when segmentation masks or contact annotations are noisy, unavailable, or less accurate.**
> >
> > **Response:** Contact labels are computed during simulation-based data collection by tracking the distance between the end-effector and objects, ensuring high ground-truth accuracy. As shown in Fig. 6, omitting the contact predictor degrades overall performance.
> > While the contact labels are highly accurate (as they are collected directly from the simulation), we conducted an ablation study where we randomly flipped the contact labels for 20% of the training samples on the Cube-Single task and find that the performance remains stable at about **72%** success rate (same as that with accurate contact labels). While we currently focus on simulation experiments, in the real world, contact labels can be generated using trajectory labeling methods (e.g., Yupu, et al., 2026).
> >
> > For object-centric learning, SAM2 masks are generated by manually prompt-pointing a single initial frame per dataset, which the SAM2 API then propagates across all frames. Typically, without the segmentation masks, small objects such as the cube are often merged together with larger objects like the robot. We show a qualitative example of this in Section F (Appendix) and confirm that omitting this mask-based loss significantly degrades object-centric representation learning.
> >
> > Lu, Yupu, Hang Xu, Yizhou Chen, and Jia Pan. "Keypose Exploration: Efficient Automatic Trajectory Labelling and Cross-Embodiment Policy Transfer." arXiv preprint arXiv:2606.29028 (2026).

---

> > > ### Author Response · Authors · 2026-07-21
> > > **Response to Reviewer xosu Part 3/n**
> > >
> > > **Concern 5: In addition, the particle-filter planning procedure can introduce significant test-time overhead. More detailed reporting of variance, runtime, and compute cost would make the empirical claims more convincing.**
> > >
> > > **Response:** We have added a detailed runtime and compute cost breakdown for all methods in Table 7 in the appendix of the revised manuscript and shown the same below. These are reported for execution on a single A4000 GPU with 16 GB of GPU memory. While WorldDP’s planning phase introduces higher test-time overhead (~65 seconds) than baselines like DP and HECRL, it yields higher success rates on multi-stage tasks. Crucially, once subgoals are planned, the policy executes actions in real-time without latency. Moreover, the planning time can be further reduced through parallel computing over several GPUs. Furthermore, under identical planning parameters, our particle-filter optimization incurs roughly the same computational runtime as standard Cross-Entropy Method (CEM) optimization.
> > >
> > > Table: Planning time
> > > | Method | Planning/Execution Time (s) |
> > > | :--- | :---: |
> > > | DinoWM | 237.3 |
> > > | LeWM | 6.9 |
> > > | HWM | 31.1 |
> > > | HWM$^\dagger$ | 360 |
> > > | HECRL$^*$ | 1.1 |
> > > | WorldDP w/o PF (with CEM) | 64.7 |
> > > | **WorldDP (Ours)** | 64.5 |

---

### Review · Reviewer_8fwb · 2026-06-24

**Summary Of Contributions:**

This paper introduces **WorldDP**, a hierarchical framework for multi-stage robotic manipulation that couples an object-centric latent world model (high-level planner) with a goal-conditioned Diffusion Policy (low-level executor). The framework has four main components: (1) an **Object-Centric Encoder (OCE)** that applies slot attention on top of frozen DINOv2 patch features, supervised during training by SAM2-generated segmentation masks via a size-weighted Tversky loss; (2) a **Conditional Diffusion Transformer (CDiT)** dynamics model that predicts future object-slot states from latent action embeddings; (3) a **two-tier MPC scheme** in which the world model optimizes high-level subgoals using a **particle filter** with an object-centric cost function augmented by a learned **contact predictor**; and (4) a goal-conditioned **Diffusion Policy** that sequentially tracks the generated subgoals. The authors evaluate on four manipulation tasks derived from OGBench (Cube-Single, Cube-Triple, Scene-Single-Direct, Scene-Single-Composite) and report consistent improvements over world-model baselines (DINO-WM, LeWM), a hierarchical baseline (HECRL), and diffusion-policy baselines (DP40, DP100), most notably on the multi-stage Cube-Triple task.

**Additional Comments:**

### Strengths

1. **Well-motivated problem and sensible high-level design.** The observation that JEPA-style world-model planners largely succeed only on single-stage tasks (reaching, grasping) and degrade on long-horizon, multi-stage manipulation is accurate and important. Decomposing the problem into world-model subgoal planning plus a fast, robust diffusion-policy tracker is a reasonable and well-argued architectural choice, and the motivation for each component (Sec. 1) is clearly developed.

2. **Strong headline results on the hardest task.** On Cube-Triple, WorldDP reaches 30% on the all-three-cubes metric versus 12% for the next-best baseline (HECRL), and 72% vs. 34% on two cubes (Table 1). This is a meaningful margin on the task most representative of the paper's multi-stage thesis.

3. **Reasonable ablation coverage.** The paper isolates the contributions of the diffusion policy (w/o DP), the object-centric encoder (w/o OCE), the particle filter vs. CEM (w/o PF), the contact predictor (w/o CP), and the object- vs. image-embedding cost function (Table 3, Figs. 5–6). This is more thorough than many comparable submissions.

4. **Adequate implementation detail.** Hyperparameters for the OCE, CDiT, diffusion policy, contact predictor, particle filter, and success-detection thresholds are documented in the appendix (Tables 4–5, Sec. A–E), which aids reproducibility.

### Weaknesses

1. **No statistical rigor; results may not be distinguishable from noise.** Evaluation uses only 50 held-out trajectories per task, and there is no indication of multiple seeds, confidence intervals, standard deviations, or significance tests anywhere in the paper. Robotic-manipulation success rates are well known to have high run-to-run variance, and several reported differences are small (e.g., the "w/o" ablations in Figs. 5–6, and individual entries in Table 2). Without variance estimates over multiple seeds, it is impossible to judge which gains are real. This is the single most important issue for a claims-must-be-supported venue like TMLR.

2. **The contribution is primarily integration, and some novelty claims are overstated.** Each component is drawn from prior work: slot attention on DINOv2 features (Locatello et al., 2020; Singh et al., 2025; Mosbach et al., 2024), the CDiT dynamics model (Goswami et al., 2025), and the diffusion policy (Chi et al., 2025). Two claims in particular need to be tempered or substantiated: "the first world model approach to employ object-aware planning," and that "the use of particle filters is new for planning using world models." Particle filtering and other sampling-based optimizers are long-standing tools in MPC/planning, and the PF here functions essentially as a multi-hypothesis replacement for CEM. The genuine contribution — a well-engineered integration that works on multi-stage tasks — is valuable, but the framing should match it.

3. **The central motivating claim about patch features is internally contradictory and not directly supported.** The paper motivates the approach by asserting that DINOv2 patch-level features "are suboptimal for state representation for dynamics learning as they often omit critical fine-grained details" (Sec. 1). However, WorldDP's object-centric representation is itself learned *directly on top of the same frozen DINOv2 patch features* — so any fine-grained detail those features discard is equally absent at WorldDP's input. The authors in fact concede this ("small objects ... may be under-represented in Eq. 1," Sec. 3.1), and Sec. 4.3 notes that the SAM2-supervised masks are downsampled to patch resolution and achieve only "patch-level, rather than pixel-level, accuracy." The method therefore adds object *grouping* and segmentation supervision, not finer spatial granularity, which contradicts the stated motivation. Moreover, the specific mechanistic claim (that patch features hurt dynamics learning by omitting fine-grained detail) is never isolated experimentally. The closest evidence, the "w/o OCE" ablation that feeds raw DINOv2 patch embeddings as states (Fig. 5), runs only on Scene-Single and **confounds** the state representation with the object-centric cost function and per-object planning — so it cannot attribute the gain to detail preservation rather than to object-wise cost/planning. The authors should either support the fine-grained-detail claim with a controlled experiment (e.g., feeding DINOv2 patch features directly to the CDiT dynamics model while holding the object-centric planning machinery fixed) or restate the motivation to reflect that the benefit comes from object decomposition for planning, not from recovering detail that their representation also lacks.

4. **The most directly comparable prior method is discussed but not evaluated.** HWM (Zhang et al., 2026) is described as "the first latent world model to optimize latent actions ... for two-level hierarchical planning" — i.e., the closest competitor to WorldDP's central claim. Yet it does not appear as an empirical baseline. The included HECRL baseline is itself a modified adaptation (RL controller replaced by the authors' own DP, single-view inputs), which makes it a weaker point of comparison than a faithful hierarchical-world-model baseline would be.

5. **Limited evaluation scope undercuts the generalization claims.** All experiments are in simulation, on a single benchmark family (OGBench variants), with four closely related tasks (cubes plus a drawer/window/button scene). The conclusion claims "superior generalization," and the paper mentions cross-embodiment learning as motivation, but there are no real-robot results, no cross-environment transfer, and no cross-embodiment experiments. The evidence supports "works well on these four simulated tasks," not broad generalization.

**Audience:**

Yes

**Audience Explanation:**

N/A

**Claims And Evidence:**

No

**Claims Explanation:**

Please check the weaknesses section.

**Requested Changes:**

1. **Report variance.** Run each configuration over multiple seeds and report means with confidence intervals or standard deviations for all main tables (1, 2) and ablations (Figs. 5–6, Table 3). Indicate which differences are statistically meaningful.

2. **Clarify and equalize the experimental setup.** State explicitly whether baselines receive the end-effector depth-camera pose correction (Sec. 3.4); if not, either extend it to baselines or report WorldDP without it (or as a separate ablation) so that gains are attributable to the proposed framework. Add a table enumerating what supervision/privileged information each method uses.

3. **Add the most relevant baseline.** Include HWM (Zhang et al., 2026), or another faithful hierarchical latent-world-model planner, as an empirical baseline, since it is the closest competitor to the paper's core claim.

4. **Temper or substantiate novelty claims.** Reframe "first object-aware world-model planning" and "particle filters are new for world-model planning" to accurately reflect prior use of sampling-based optimizers and object-centric planning; position the contribution as a strong integration.

5. **Resolve the patch-feature contradiction.** Either (a) provide a controlled experiment that feeds DINOv2 patch features directly to the CDiT dynamics model while keeping the object-centric planning/cost machinery fixed, to isolate whether the gain comes from the state representation or from object-wise planning; or (b) restate the motivation so it no longer claims to address "omitted fine-grained details" — which the patch-resolution OCE and SAM2 masks do not recover — and instead attributes the benefit to object decomposition for planning.

---

> ### Author Response · Authors · 2026-07-21
> **Response to Reviewer 8fwb Part 1/n**
>
> We thank the reviewer for the detailed review and for the constructive comments. We further appreciate your recognition of the strengths of the paper. In the following, we address each of the concerns (weaknesses and requested changes).
>
> **Concern 1: No statistical rigor; results may not be distinguishable from noise.**
>
> **Response:** We have updated Tables 1, 2, and 3, and Figures 5 and 6 to report the mean and standard deviation across 10 seeds, confirming that WorldDP's performance gains over the established baselines are statistically significant. The results from Tables 1 and 2 are shown below for reference. These results follow the original pattern: WorldDP consistently outperforms baselines and shows the highest consistency, even on complex tasks like Cube-Triple. Furthermore, the new generalization experiments in Section 4 (also shown in response to your Concern 6) show that WorldDP maintains meaningful advantages over baselines under adverse environmental perturbations (changing object and background colors, camera angles, and lighting).
>
> Table 1: Success rate (%) on Cube-Triple and Scene-Single-Composite. 1-Cube, 2-Cubes, and 3-Cubes in Cube-Triple refer to at least 1, 2, and all 3 cube successes. Mean and Standard Deviation are reported over 10 seeds.
> | Method | Cube-Triple (1-Cube) | Cube-Triple (2-Cubes) | Cube-Triple (3-Cubes) | Scene-Single-Composite (Button Press) | Scene-Single-Composite (Full Task) |
> | :--- | :---: | :---: | :---: | :---: | :---: |
> | DinoWM | 60.4±4.6 | 11.8±4.3 | 0.8±1.0 | 3.4±2.1 | 0.2±0.6 |
> | LeWM | 67.0±5.1 | 10.4±2.1 | 0.0±0.0 | 4.0±2.5 | 0.2±0.6 |
> | HWM | 56.4±5.4 | 7.0±2.9 | 0.6±1.0 | 1.8±2.0 | 0.6±1.3 |
> | HWM$^\dagger$ | 61.2±5.1 | 9.4±1.9 | 0.4±0.8 | 24.2±6.8 | 8.2±3.9 |
> | HECRL$^*$ | 75.2±5.6 | 33.0±7.7 | 4.8±3.7 | 55.4±4.8 | 19.0±4.1 |
> | DP40 | 65.4±1.3 | 9.4±1.0 | 0.0±0.0 | **69.4±5.2** | 0.0±0.0 |
> | DP100 | 76.2±3.5 | 30.4±5.1 | 2.2±1.1 | 63.8±6.3 | 9.2±2.9 |
> | **WorldDP (Ours)** | **92.8±2.5** | **70.6±5.0** | **29.4±4.6** | 51.0±10.2 | **20.2±5.4** |
>
>
> Table 2: Success rate (%) on Cube-Single and Scene-Single-Direct tasks. Mean and Standard Deviation are reported over 10 seeds.
> | Method | Cube-Single | Scene-Single-Direct (Button) | Scene-Single-Direct (Drawer) | Scene-Single-Direct (Window) | Scene-Single-Direct (Average) | Both Task Avg |
> | :--- | :---: | :---: | :---: | :---: | :---: | :---: |
> | DinoWM | 0.0±0.0 | 16.2±9.4 | 25.9±9.3 | 19.4±10.0 | 20.5±4.3 | 10.3±2.2 |
> | LeWM | 0.4±1.3 | 16.9±4.2 | 4.1±5.6 | 19.4±8.3 | 13.5±2.8 | 6.9±1.7 |
> | HWM | 1.0±1.4 | 0.6±2.0 | 8.8±6.9 | 3.5±4.1 | 4.3±2.6 | 2.7±1.5 |
> | HWM$^\dagger$ | 0.2±0.6 | 43.1±15.2 | 48.8±8.3 | 21.2±6.9 | 37.7±6.7 | 19.0±3.3 |
> | HECRL$^*$ | **99.0±1.4** | 50.0±8.8 | 48.8±10.8 | 8.8±3.1 | 35.9±4.3 | 67.4±2.2 |
> | DP40 | 0.0±0.0 | **81.2±6.6** | 27.6±4.0 | 6.5±3.3 | 38.5±1.9 | 19.2±0.9 |
> | DP100 | 98.0±0.9 | 66.9±7.8 | 21.2±5.0 | 4.7±3.7 | 30.9±2.8 | 64.5±1.3 |
> | **WorldDP (Ours)** | 71.2±6.6 | 65.6±13.6 | **75.3±8.2** | **63.5±11.4** | **68.1±7.3** | **69.7±4.0** |
>
> **Concern 2: Clarify and equalize the experimental setup. State explicitly whether baselines receive the end-effector depth-camera pose correction (Sec. 3.4); if not, either extend it to baselines or report WorldDP without it (or as a separate ablation) so that gains are attributable to the proposed framework. Add a table enumerating what supervision/privileged information each method uses.**
>
> **Response:** The end-effector depth-camera pose correction is integrated directly into the low-level Diffusion Policy pipeline for the Cube-Single and Cube-Triple tasks. Consequently, the Diffusion Policy baselines (DP40, DP100) and HECRL* (which uses Diffusion Policy in its lower hierarchy) receive this exact same correction. The DinoWM, LeWM, and HWM baselines do not use Diffusion Policy in any capacity and thus do not receive it. Crucially, WorldDP still outperforms all baselines in the Scene-Single-Direct and Scene-Single-Composite tasks on average, where the depth-camera correction is not used at all. We have clarified this setup in the manuscript and added Table 8 in the Appendix (also shown below) enumerating the supervision used by each method. Notably, because the Diffusion Policy use the same object-centric state representation as WorldDP (Sec. 3.4), baselines HECRL*, DP40, and DP100 also utilize segmentation masks to train their state encoder.
>
> | Method | Training <br> Expert Trajectories | Training <br> Segmentation Masks (SAM2) | Training <br> Contact Labels | Evaluation <br> Depth-Camera Pose Correction |
> | :--- | :---: | :---: | :---: | :---: |
> | DinoWM | ✓ | | | |
> | LeWM | ✓ | | | |
> | HWM | ✓ | | | |
> | HWM$^\dagger$ | ✓ | ✓ | ✓ | |
> | HECRL$^*$ | ✓ | ✓ | | ✓ (Cube tasks only) |
> | DP40 / DP100 | ✓ | ✓ | | ✓ (Cube tasks only) |
> | **WorldDP (Ours)** | **✓** | **✓** | **✓** | **✓ (Cube tasks only)** |

---

> > ### Author Response · Authors · 2026-07-21
> > **Response to Reviewer 8fwb Part 2/n**
> >
> > **Concern 3: The most directly comparable prior method, HWM (Zhang et al., 2026), is discussed but not evaluated.**
> >
> > **Response:** We have included HWM (Zhang et al., 2026) in Tables 1 and 2 of the updated manuscript (also shown above in response to your Concern 1), evaluating two versions. Since the open-source HWM code is released only for smaller-scale maze environments, we implemented both a direct, faithful baseline (HWM) and a scaled version ($\text{HWM}^\dagger$). For $\text{HWM}^\dagger$ we used WorldDP’s exact high-level world model architecture, weights, and high-level planning framework, but replaced the low-level policy with a 1-step transition model that plans explicitly for actions to reach subgoals, same as HWM. The results demonstrate that WorldDP outperforms both HWM variants across all tasks. Further, we note that the performance of $\text{HWM}^\dagger$, which includes WorldDP’s high-level model, is better than that of HWM.
> >
> > **Concern 4: The contribution is primarily integration, and some novelty claims are overstated.**
> >
> > **Response:** WorldDP is the first hierarchical latent space (JEPA) world model where subgoals are generated in the top hierarchy via particle-filter-based planning on object-centric representations, and reached via a goal-conditioned diffusion policy in the lower hierarchy. While latent-space world models (e.g., Zhou et al., 2024, Goswami et al., 2025), object-centric representations (e.g., Singh et al., 2025), and Diffusion Policy (Chi et al., 2025) are existing individual paradigms, integrating them into a unified framework for multi-stage tasks requires novel framework-level contributions: (a) SAM2-supervised object representation learning; (b) extending CDiT  from Goswami et al., 2025 to use object embeddings and an action encoder; (c) sequential object-wise particle-filter planning; and (d) integrating a contact predictor during planning.
> >
> > **"the first world model approach to employ object-aware planning"**
> >
> > WorldDP is the first to use object-centric world models as system dynamics functions within an MPC framework to plan for actions. Most existing object-centric world models (e.g., Ferraro et al., 2023) either predict actions directly or rely on RL-based frameworks. We have clarified this distinction in Section 2 of the updated manuscript.
> >
> > **"the use of particle filters is new for planning using world models"**
> >
> > Existing learned world models that optimize actions at runtime (e.g., Zhou et al., 2024; Goswami et al., 2025) rely on CEM. WorldDP is the first in this line of work to employ a multi-hypothesis particle filter instead. We have clarified this in Section 3.3 of the updated manuscript.
> >
> > **Concern 5: The central motivating claim about patch features is internally contradictory and not directly supported.**
> >
> > **Response:** In the original version of the paper, when we wrote that "patch-level features are suboptimal for state representation for dynamics learning as they often omit critical fine-grained details," we were referring to what happens when patch-level features are directly used as system states, rather than stating generally that DINOv2 features themselves lack fine-grained details. Indeed, in such setups, dynamics models tend to over-focus on dominant background and robot patches, minimizing signals from small objects (e.g., cubes) that occupy very few patches, effectively omitting them from the dynamics model's predictions. We agree that DINOv2 patch-level features must contain information about small objects for the object-centric encoder to extract them. To avoid confusion, we have removed claims about addressing "omitted fine-grained details" from the updated manuscript and reworded the text for better clarity.
> >
> > Similarly, when we state that small objects occupying few pixels may be under-represented in Eq. 1, we refer to the fact that small objects take up a significantly lower number of patches compared to larger entities like the robot or background. Transitioning to an object-centric representation forces the dynamics model to weight all scene entities equitably and enables sequential planning per object during evaluation.
> >
> > We have revised the manuscript to accurately reflect that our performance gains stem from object decomposition for state prediction and entity-focused planning. Specifically, we updated the introduction and related work to clarify these points, and further clarified Eq. 1 in the revised text.
> >
> >
> > **References**
> >
> > Ferraro, Stefano, Pietro Mazzaglia, Tim Verbelen, and Bart Dhoedt. "FOCUS: object-centric world models for robotic manipulation." Frontiers in Neurorobotics 19 (2025): 1585386.

---

> ### Author Response · Authors · 2026-07-21
> **Response to Reviewer 8fwb Part 3/n**
>
> **Concern 6: Limited evaluation scope undercuts the generalization claims.**
>
> **Respone:** We evaluate all tasks on held-out test sets from the OGBench suite containing unseen variations, where WorldDP consistently outperforms established baselines.  OGBench is a standard benchmark in recent literature(e.g., Haramati et al., 2026, Maes et al., 2026b) and tests diverse skills, including pick-and-place, articulated object manipulation (e.g., drawers/windows), and physical reasoning, posing significant challenges to existing baselines such as DinoWM, LeWM, HWM, and HECRL\*.  To further validate our generalization claims, we have added new generalization experiments in Section 4 (also shown below) where we compared WorldDP against the top-performing baselines (HECRL\*, DP40, DP100) across 10 episodes over 3 seeds for each task under adverse environmental perturbations (changing object and background colors, camera angles, and lighting). Across Cube-Triple, Scene-Single-Composite, and Scene-Single-Direct, WorldDP outperforms all baselines on almost every condition. On Cube-Single, it retains higher relative robustness under adverse environment shifts than established baselines like HECRL\*. We have also revised the conclusion to clarify that our generalization here refers to robustness under such environmental variations. Further, as already mentioned in the conclusion, we leave tasks like cross-embodiment learning for future work.
>
> Table: Generalization to unseen environment conditions: Camera Viewpoint Perturbations; Random Floor Color; Random Lighting; and Random Object Color.
> | Task Setting | Method | Default | Camera | Floor | Lighting | Object |
> | :--- | :--- | :---: | :---: | :---: | :---: | :---: |
> | **Cube-Triple** | DP40 | 0.0 ± 0.0 | 0.0 ± 0.0 | 0.0 ± 0.0 | 0.0 ± 0.0 | 0.0 ± 0.0 |
> | | DP100 | 0.0 ± 0.0 | 3.3 ± 5.8 | 0.0 ± 0.0 | 0.0 ± 0.0 | 0.0 ± 0.0 |
> | | HECRL$^*$ | 0.0 ± 0.0 | 10.0 ± 10.0 | 3.3 ± 5.8 | 13.3 ± 15.3 | **3.3 ± 5.8** |
> | | **WorldDP (Ours)** | **20.0 ± 17.3** | **13.3 ± 15.3** | **6.7 ± 5.8** | **30.0 ± 10.0** | 0.0 ± 0.0 |
> | | | | | | | |
> | **Scene-Single-Composite** | DP40 | 0.0 ± 0.0 | 0.0 ± 0.0 | 0.0 ± 0.0 | 0.0 ± 0.0 | 0.0 ± 0.0 |
> | | DP100 | 10.0 ± 17.3 | 0.0 ± 0.0 | **3.3 ± 5.8** | 6.7 ± 11.6 | 3.3 ± 5.8 |
> | | HECRL$^*$ | **40.0 ± 0.0** | **3.3 ± 5.8** | 0.0 ± 0.0 | 10.0 ± 10.0 | 10.0 ± 10.0 |
> | | **WorldDP (Ours)** | 10.0 ± 10.0 | **3.3 ± 5.8** | 0.0 ± 0.0 | **16.7 ± 20.8** | **20.0 ± 10.0** |
> | | | | | | | |
> | **Cube-Single** | DP40 | 0.0 ± 0.0 | 0.0 ± 0.0 | 0.0 ± 0.0 | 0.0 ± 0.0 | 0.0 ± 0.0 |
> | | DP100 | **100.0 ± 0.0** | **80.0 ± 10.0** | **83.3 ± 5.8** | **100.0 ± 0.0** | **100.0 ± 0.0** |
> | | HECRL$^*$ | **100.0 ± 0.0** | **80.0 ± 10.0** | 23.3 ± 15.3 | 93.3 ± 5.8 | 53.3 ± 25.2 |
> | | **WorldDP (Ours)** | 83.3 ± 5.8 | 66.7 ± 15.3 | 46.7 ± 11.6 | 76.7 ± 23.1 | 76.7 ± 5.8 |
> | | | | | | | |
> | **Scene-Single-Direct** | DP40 | 40.0 ± 10.0 | 13.3 ± 5.8 | 3.3 ± 5.8 | 40.0 ± 10.0 | 13.3 ± 5.8 |
> | | DP100 | 36.7 ± 5.8 | 20.0 ± 17.3 | 0.0 ± 0.0 | 36.7 ± 5.8 | 10.0 ± 0.0 |
> | | HECRL$^*$ | 40.0 ± 0.0 | 16.7 ± 20.8 | 10.0 ± 10.0 | 23.3 ± 5.8 | 20.0 ± 17.3 |
> | | **WorldDP (Ours)** | **66.7 ± 15.3** | **50.0 ± 10.0** | **40.0 ± 17.3** | **66.7 ± 20.8** | **63.3 ± 5.8** |

---

### Review · Reviewer_bkjx · 2026-07-11

**Summary Of Contributions:**

The paper proposes WorldDP, which is a hierarchical framework for multi-stage robotic manipulation conditioned on target images. The model is trained on reward-free play data. The main contribution is the architectural design of the hierarchical pipeline. A CDiT latent dynamics model is trained on per-entity slot states produced by an object-centric encoder. The dynamics model is used inside a sampling-based optimizer with a slot-level MSE cost plus a learned contact predictor cross-entropy term to produce subgoals for a lower-level, goal-conditioned diffusion policy that tracks the subgoals. Experiments are conducted on four tasks derived of OGBench, and ablations are provided for all the core elements of the algorithm.

**Audience:**

Yes

**Audience Explanation:**

Long-horizon and multi-stage robotic robotic manipulation is an active area of research. The interplay between JEPA-style latent world-model planning and diffusion policies is an active direction. The paper presents findings that would be of genuine interest to researchers in world models and robot learning, such as the finding that diffusion policy is a more robust low-level tracker than direct action optimization together with the observation that CEM's unimodal search underperforms a multi-seed sampler on multimodal manipulation.

**Broader Impact Concerns:**

I have no broader impact concerns beyond what was already discussed.

**Claims And Evidence:**

No

**Claims Explanation:**

The algorithm is well explained, the authors ablate every proposed ingredient, but the evidence has gaps that prevent the claims from being adequately supported yet. I identify four main limitations of the work:

1. Variance is not reported for any of the experiments.
2. The authors position WorldDP directly against HWM ("instead of using another world model at the lower level, we use a diffusion policy"), but there is no empirical comparison against that baseline. Also, how does HWM compare against the "w/o DP" ablation in Fig. 5?
3. The component ablations are well designed to cover every proposed ingredient, but they are run on disjoint task subsets with no full task vs. component matrix, and it is unclear whether the CEM used in "w/o PF" receives the same rollout budget as the PF.
4. Some novelty claims are overstated. E.g., "the use of particle filters is new for planning using world models" (particle-filter and sequential-Monte-Carlo MPC are well established for sampling-based control [1, 2]) or "We present the first world model approach to employ object-aware planning" (the authors themselves cite LPWM (Daniel et al. 2026) and SOLD (Mosbach et al. 2024), both object-centric world models). Moreover, the procedure described in Algorithm 1 (multiple seed means -> Gaussian sampling -> top-M elites -> iterate) has no sequential importance weighting or resampling against an observation likelihood, so it deviates from standard particle filtering.

Beyond that, there are some necessary clarifications that I detail in the "Requested Changes".

[1] D. Stahl and J. Hauth, "PF-MPC: Particle filter–model predictive control", Systems & Control Letters, 60(8):632–643, 2011.

[2] N. Kantas, J. M. Maciejowski, A. Lecchini-Visintini, "Sequential Monte Carlo for Model Predictive Control", 2009.

**Requested Changes:**

Critical changes necessary for the claims to be supported by evidence:

1. The empirical findings are a big part of the contribution, so please report variance.
2. Compare to HWM, or justify its absence concretely. Even a comparison on a subset of tasks or an ablation run in HWM's style would substantiate the paper's central architectural argument.
3. Run the key component ablations (w/o PF, w/o CP, w/o OCE) on a shared task set, or justify the current subset choices.
4. Scope the novelty claims mentioned earlier (the use of particle filters for planning using world models, and the use of object-aware planning).
5. Please clarify supervision and evaluation parity with the baselines:

    i. How are the per-entity ground-truth masks for SAM2 supervision obtained?

    ii. How are contact-predictor training labels obtained?

    iii. Is the depth-camera residual grasp correction applied to DP40/DP100 and HECRL* as well?
6. Particle filtering can incur a significant cost at inference time. Please, report wall-clock time for all methods and confirm that the "w/o PF" (CEM) ablation is compute-matched to the PF, otherwise that ablation conflates optimizer quality with sampling budget.
7. Please, clarify the provenance of the values of the column "Both Task Average" in Table 2, the values don't follow from the other columns under averaging.

Improvements:

8. It would be interesting to study object-centric failure modes, such as occlusion, deformable objects, objects that split (e.g., Lego), and scenes with more or fewer entities than the fixed N slots. The paper never discusses any of these, which are very relevant for real-world applicability.
9. The motivation says patch features "omit critical fine-grained details", but a state consisting of N d-dimensional slots is likely more compressed than P d-dimensional patches (with N usually much smaller than P). That motivation argues against slots too, so the motivation could be adjusted.
10. It is stated that diffusion policy "can compensate for suboptimal subgoals to a good extent". It would be interesting to elaborate on that point, for example, in the "w/o DP" ablation.
11. Algorithm 1 could be clarified to better reflect the MPC structure. As written, it shows a single planning call followed by open-loop subgoal tracking. Add the outer loop and the replanning granularity (per object).

---

> ### Author Response · Authors · 2026-07-21
> **Response to Reviewer bkjx Part 1/n**
>
> We thank the reviewer for the detailed review and for the constructive comments. We further appreciate your recognition of the strengths of the paper. In the following, we address each of the concerns (weaknesses and requested changes).
>
> **Concern 1: Variance is not reported for any of the experiments.**
>
> **Response:** All experimental results in the paper have been updated to report the mean and standard deviation across 10 seeds. The results from Tables 1 and 2 are also shown below. The results follow the original pattern: WorldDP consistently outperforms baselines and shows the highest consistency, even on complex tasks like Cube-Triple.
>
> Table 1: Success rate (%) on Cube-Triple and Scene-Single-Composite. 1-Cube, 2-Cubes, and 3-Cubes in Cube-Triple refer to at least 1, 2, and all 3 cube successes. Mean and Standard Deviation are reported over 10 seeds.
> | Method | Cube-Triple (1-Cube) | Cube-Triple (2-Cubes) | Cube-Triple (3-Cubes) | Scene-Single-Composite (Button Press) | Scene-Single-Composite (Full Task) |
> | :--- | :---: | :---: | :---: | :---: | :---: |
> | DinoWM | 60.4±4.6 | 11.8±4.3 | 0.8±1.0 | 3.4±2.1 | 0.2±0.6 |
> | LeWM | 67.0±5.1 | 10.4±2.1 | 0.0±0.0 | 4.0±2.5 | 0.2±0.6 |
> | HWM | 56.4±5.4 | 7.0±2.9 | 0.6±1.0 | 1.8±2.0 | 0.6±1.3 |
> | HWM$^\dagger$ | 61.2±5.1 | 9.4±1.9 | 0.4±0.8 | 24.2±6.8 | 8.2±3.9 |
> | HECRL$^*$ | 75.2±5.6 | 33.0±7.7 | 4.8±3.7 | 55.4±4.8 | 19.0±4.1 |
> | DP40 | 65.4±1.3 | 9.4±1.0 | 0.0±0.0 | **69.4±5.2** | 0.0±0.0 |
> | DP100 | 76.2±3.5 | 30.4±5.1 | 2.2±1.1 | 63.8±6.3 | 9.2±2.9 |
> | **WorldDP (Ours)** | **92.8±2.5** | **70.6±5.0** | **29.4±4.6** | 51.0±10.2 | **20.2±5.4** |
>
>
> Table 2: Success rate (%) on Cube-Single and Scene-Single-Direct tasks. Mean and Standard Deviation are reported over 10 seeds.
> | Method | Cube-Single | Scene-Single-Direct (Button) | Scene-Single-Direct (Drawer) | Scene-Single-Direct (Window) | Scene-Single-Direct (Average) | Both Task Avg |
> | :--- | :---: | :---: | :---: | :---: | :---: | :---: |
> | DinoWM | 0.0±0.0 | 16.2±9.4 | 25.9±9.3 | 19.4±10.0 | 20.5±4.3 | 10.3±2.2 |
> | LeWM | 0.4±1.3 | 16.9±4.2 | 4.1±5.6 | 19.4±8.3 | 13.5±2.8 | 6.9±1.7 |
> | HWM | 1.0±1.4 | 0.6±2.0 | 8.8±6.9 | 3.5±4.1 | 4.3±2.6 | 2.7±1.5 |
> | HWM$^\dagger$ | 0.2±0.6 | 43.1±15.2 | 48.8±8.3 | 21.2±6.9 | 37.7±6.7 | 19.0±3.3 |
> | HECRL$^*$ | **99.0±1.4** | 50.0±8.8 | 48.8±10.8 | 8.8±3.1 | 35.9±4.3 | 67.4±2.2 |
> | DP40 | 0.0±0.0 | **81.2±6.6** | 27.6±4.0 | 6.5±3.3 | 38.5±1.9 | 19.2±0.9 |
> | DP100 | 98.0±0.9 | 66.9±7.8 | 21.2±5.0 | 4.7±3.7 | 30.9±2.8 | 64.5±1.3 |
> | **WorldDP (Ours)** | 71.2±6.6 | 65.6±13.6 | **75.3±8.2** | **63.5±11.4** | **68.1±7.3** | **69.7±4.0** |
>
> **Concern 2: Compare to HWM, or justify its absence concretely. Also, how does HWM compare against the "w/o DP" ablation in Fig. 5?**
>
> **Response:** We have added HWM baseline in the updated manuscript (also shown in the response to your Concern 1). Since the open-source HWM code is released only for smaller-scale maze environments, we implemented both a direct, faithful baseline (HWM) and a scaled version ($\text{HWM}^\dagger$). For $\text{HWM}^\dagger$, we used WorldDP’s exact high-level world model framework, weights, and high-level planning, but replaced the low-level policy with a 1-step transition model that plans explicitly for actions to reach subgoals, same as HWM. WorldDP consistently outperforms both variants across all tasks. Further, we note that the performance of $\text{HWM}^\dagger$, which includes WorldDP’s high-level model, is significantly better than that of HWM.
>
> In the "w/o DP" ablation, the optimization variables are low-level physical actions applied directly to the robot, rather than the high-level latent actions optimized in WorldDP and HWM's upper hierarchy. In contrast, HWM uses a second lower-level optimization to map physical actions toward each high-level subgoal, while WorldDP predicts these low-level actions using its diffusion policy.
>
> **Concern 3: The component ablations are well designed to cover every proposed ingredient, but they are run on disjoint task subsets with no full task vs. component matrix, and it is unclear whether the CEM used in "w/o PF" receives the same rollout budget as the PF.  Run the key component ablations (w/o PF, w/o CP, w/o OCE) on a shared task set, or justify the current subset choices.**
>
> **Response:** Although the "w/o PF" and "w/o CP" ablations are reported in Fig. 6, and the "w/o OCE" and "w/o DP" ablations are reported in Fig. 5, all of these ablations were actually run on the same shared task set: Scene-Single-Direct and Scene-Single-Composite as already noted in the paper. These tasks evaluate both direct (opening/closing drawers, windows, and pressing buttons) and composite capabilities. The CEM used in the "w/o PF" ablation receives the exact same rollout budget as the PF.

---

> > ### Author Response · Authors · 2026-07-21
> > **Response to Reviewer bkjx Part 2/n**
> >
> > **Concern 4: Scope the novelty claims mentioned earlier (the use of particle filters for planning using world models, and the use of object-aware planning).**
> >
> > **Response:**
> >
> > **"the use of particle filters is new for planning using world models"**
> >
> > Existing learned world models that optimize actions at runtime (such as Zhou et al., 2024 and Goswami et al., 2025) rely on CEM. WorldDP is the first in this line of work to employ a multi-hypothesis particle filter instead. We have tempered this claim to clarify this specific distinction.
> >
> > **"We present the first world model approach to employ object-aware planning"**
> >
> > WorldDP is the first to use object-centric world models as system dynamics functions within an MPC framework to plan for actions. Most existing object-centric world models either predict actions directly or rely on RL policies. For example, LPWM predicts latent actions directly using a latent inverse dynamics model rather than optimizing them via runtime planning, while SOLD relies on an Actor-Critic framework for action prediction. We have clarified these differences in the paper.
> >
> > **Concern 5: Please clarify supervision and evaluation parity with the baselines:
> > i. How are the per-entity ground-truth masks for SAM2 supervision obtained?
> > ii. How are contact-predictor training labels obtained?
> > iii. Is the depth-camera residual grasp correction applied to DP40/DP100 and HECRL\* as well?**
> >
> > **Response:** i. SAM2 masks are generated by manually prompt-pointing a single initial frame per dataset, which the SAM2 API then propagates across all frames.
> > ii. Contact labels are computed during simulation-based data collection by tracking the distance between the end-effector and objects.
> > iii. Yes, the depth-camera residual grasp correction is also applied to DP40, DP100, and HECRL\* .
> >
> > **Concern 6: Particle filtering can incur a significant cost at inference time. Please, report wall-clock time for all methods and confirm that the "w/o PF" (CEM) ablation is compute-matched to the PF, otherwise that ablation conflates optimizer quality with sampling budget.**
> >
> > **Response:** We have added a detailed runtime cost breakdown for all methods in Table 7 of the revised manuscript and shown the same below. These experiments are conducted on a single A4000 GPU with 16 GB of GPU memory. While WorldDP’s planning phase introduces higher test-time overhead (~65 seconds) than baselines like DP and HECRL, it yields higher success rates on multi-stage tasks. Crucially, once subgoals are planned, the policy executes actions in real-time without latency. Moreover, the planning time can be further reduced through parallel computing over several GPUs. Furthermore, under identical planning parameters, our particle-filter optimization incurs roughly the same computational runtime as standard Cross-Entropy Method (CEM) optimization.
> >
> > Table: Planning time
> > | Method | Planning/Execution Time (s) |
> > | :--- | :---: |
> > | DinoWM | 237.3 |
> > | LeWM | 6.9 |
> > | HWM | 31.1 |
> > | HWM$^\dagger$ | 360 |
> > | HECRL$^*$ | 1.1 |
> > | WorldDP w/o PF (with CEM) | 64.7 |
> > | **WorldDP (Ours)** | 64.5 |
> >
> > **Concern 7: Please, clarify the provenance of the values of the column "Both Task Average" in Table 2, the values don't follow from the other columns under averaging.**
> >
> > **Response:** Thank you for pointing this out. The "Both Task Average" in Table 2 is the average of the success rates on Cube-Single and the average success rate on Scene-Single-Direct. Some values in this column were off by a few points as they were carried over from a prior draft. We apologize for the confusion. Even with the corrected values, the overall takeaways remain unchanged. We have updated the table to reflect the correct averages along with mean and standard deviation statistics in the revised manuscript.
> >
> > **Concern 8: It would be interesting to study object-centric failure modes, such as occlusion, deformable objects, objects that split (e.g., Lego), and scenes with more or fewer entities than the fixed N slots. The paper never discusses any of these, which are very relevant for real-world applicability.**
> >
> > **Response:** The primary goal of WorldDP is to use world-model-based planning for multi-stage tasks. As such, we proposed a hierarchical framework consisting of an object-centric world model and a diffusion policy, outperforming existing methods in multi-stage tasks. While we agree that handling occlusions, deformable objects, or splitting entities are relevant, these are beyond our current scope and we have added them as directions for future work. For instance, occlusions can be mitigated using multi-view camera setups. For deformable or splitting objects, or cases where the entity count differs from our fixed $N$ slots, a potential solution is to set a sufficiently large $N$ and focus planning on the slots showing the most significant changes between the start and goal frames. We have added a brief discussion of this in Section 5 of the updated manuscript.

---

> > > ### Author Response · Authors · 2026-07-21
> > > **Response to Reviewer bkjx Part 3/n**
> > >
> > > **Concern 9: The motivation says patch features "omit critical fine-grained details", but a state consisting of N d-dimensional slots is likely more compressed than P d-dimensional patches (with N usually much smaller than P). That motivation argues against slots too, so the motivation could be adjusted.**
> > >
> > > **Response:** In the original version of the paper, when we wrote that "patch-level features are suboptimal for state representation for dynamics learning as they often omit critical fine-grained details," we were referring to what happens when patch-level features are directly used as system states, rather than stating generally that DINOv2 features themselves lack fine-grained details. Indeed, in such setups, dynamics models tend to over-focus on dominant background and robot patches, minimizing signals from small objects (e.g., cubes) that occupy very few patches, effectively omitting them from the dynamics model's predictions. We agree that DINOv2 patch-level features must contain information about small objects for the object-centric encoder to extract them. To avoid confusion, we have removed claims about addressing "omitted fine-grained details" from the updated manuscript and reworded the text for better clarity.
> > >
> > > Transitioning to an object-centric representation forces the dynamics model to weight all scene entities equitably and enables sequential planning per object during evaluation. We have revised the manuscript to accurately reflect that our performance gains stem from object decomposition for state prediction and entity-focused planning. Specifically, we updated the introduction and related work to clarify these points, and further clarified Eq. 1 in the revised text.
> > >
> > > **Concern 10: It is stated that diffusion policy "can compensate for suboptimal subgoals to a good extent". It would be interesting to elaborate on that point, for example, in the "w/o DP" ablation.**
> > >
> > > **Response:** We have added a brief discussion of this point in Section 4.2 of the updated manuscript. Ideally, subgoals should lie on keyframes (e.g., the moment where the drawer is grasped), but in practice, they may be slightly imprecise (e.g., showing the drawer already pull a little). The diffusion policy can successfully execute the transition to reach these suboptimal targets. In contrast, directly optimizing for actions ("w/o DP" ablation) lacks the precision to reliably complete the manipulation under imprecise subgoals.
> > >
> > > **Concern 11: Moreover, the procedure described in Algorithm 1 (multiple seed means -> Gaussian sampling -> top-M elites -> iterate) has no sequential importance weighting or resampling against an observation likelihood, so it deviates from standard particle filtering. Algorithm 1 could be clarified to better reflect the MPC structure. As written, it shows a single planning call followed by open-loop subgoal tracking. Add the outer loop and the replanning granularity (per object).**
> > >
> > > **Response:** Our planning procedure is a specialized Particle-Filter-based MPC framework. Instead of updating particle weights step-by-step via sequential importance weighting, we evaluate the entire trajectory sequence at once using our dynamics model and object-centric cost. Furthermore, we replace probabilistic resampling with a deterministic step, selecting the top elite candidates to seed the next generation of Gaussian-perturbed particles. We have clarified this in the paper and further added the outer loop and replanning granularity in the Algorithm block per your suggestion.

---

### Review · Reviewer_Sxfb · 2026-07-14

**Summary Of Contributions:**

To tackle the challenges of long-horizon, sequential robotic manipulation, the authors present WorldDP, a two-tier framework that divides the control problem into high-level latent trajectory planning and low-level reactive execution. The system's state space is defined by an Object-Centric Encoder (OCE) that groups frozen DINOv2 patch features into entity-specific slot embeddings, utilizing SAM2 segmentation masks as auxiliary training targets. Transitions over these slots are modeled by a Conditional Diffusion Transformer (CDiT). For online execution, a Particle Filter optimizer generates intermediate subgoals by minimizing a composite cost function consisting of an object-distance metric and a learned contact-prediction signal. These optimized subgoals are then tracked and executed sequentially by a goal-conditioned Diffusion Policy. Evaluated across four manipulation tasks on an OGBench variant, WorldDP shows notable performance improvements over flat world models and pure imitation learning baselines.


### Strengths
- Combining high-level, physically grounded latent world models with a fast, robust diffusion policy tracker is a highly effective way to scale world models to multi-stage tasks.
- The framework achieves a notable performance margin on the challenging Cube-Triple task, which requires sequential multi-object manipulation.
- The paper provides a clear picture of how individual modules (such as the contact predictor, particle filter, and slot attention encoder) affect performance.

### Weaknesses
- A core conceptual contradiction exists in the paper's motivation regarding the quality of the state representation: while the authors assert in Section 1 that frozen DINOv2 patch-level features are suboptimal because they omit critical fine-grained details, the proposed Object-Centric Encoder (OCE) is built directly on top of these exact same frozen DINOv2 patch features, meaning the slot-attention mapping cannot mathematically recover information already discarded by the backbone. Rather than claiming to recover lost fine-grained spatial details, the authors should reframe their motivation to focus on the benefits of object-level grouping and decomposition for structured, entity-specific MPC planning.
- While the individual subtask success rates average cleanly to the reported "Scene-Single-Direct Average" for most configurations, the final "Both Task Average" values reported for HECRL, DP100, and WorldDP do not mathematically align with the simple average of their respective Cube-Single and Scene-Single-Direct scores. The authors should clarify whether a specific weighted averaging scheme was applied across tasks or correct these inconsistencies to ensure empirical consistency.

- Most components, including Slot Attention, CDiT, Diffusion Policy, Particle Filter planning, and SAM-guided object representations—already exist in prior work. The main contribution is combining these existing components into a unified framework instead of introducing fundamentally new algorithms.

- The framework assumes a fixed number of object slots for each environment. It is unclear how the method would generalize to scenes containing unknown or varying numbers of objects.

**Audience:**

Yes

**Audience Explanation:**

Despite moderate concerns regarding novelty, I believe many researchers interested in world models, robot learning, hierarchical planning, and diffusion-based control would find the empirical findings and system design informative.

**Claims And Evidence:**

No

**Claims Explanation:**

The empirical evidence generally supports the main claim that the proposed framework achieves better task performance on the evaluated benchmarks. However, several stronger claims regarding the underlying reasons for these improvements are not fully substantiated by the provided evidence.
- The paper mainly demonstrates downstream task improvements but does not directly evaluate whether the learned object-centric representations produce more accurate latent dynamics or better long-horizon predictions than patch-level representations.
- The paper claims that the proposed framework is suitable for multi-stage robotic manipulation, but the evaluation is limited to a single benchmark family (OGBench).
- There is no quantitative evaluation of rollout quality, long-horizon prediction accuracy, or latent representation fidelity, making it difficult to verify whether the proposed world model is fundamentally better or whether the gains primarily arise from the hierarchical execution framework.

**Requested Changes:**

- HECRL is modified by replacing its original low-level RL controller with the authors' diffusion policy. Since the baseline no longer matches the original algorithm, it becomes difficult to attribute performance differences solely to the proposed method. Additional discussion regarding fairness would strengthen the experimental comparison.
- Section 3.4 details that a depth-camera-mounted pose correction is utilized during execution to handle grasp errors. It is unclear if the baselines (such as HECRL or DP100) were also granted this auxiliary correction mechanism. If they were not, the performance gains cannot be cleanly attributed to the WorldDP hierarchical framework.

---

> ### Author Response · Authors · 2026-07-21
> **Response to Reviewer Sxfb Part 1/n**
>
> We thank the reviewer for the detailed review and for the constructive comments. We further appreciate your recognition of the strengths of the paper. In the following, we address each of the concerns (weaknesses and requested changes).
>
> **Concern 1: A core conceptual contradiction exists in the paper's motivation regarding the quality of the state representation: while ... entity-specific MPC planning.**
>
> **Response:** In the original version of the paper, when we wrote that "patch-level features are suboptimal for state representation for dynamics learning as they often omit critical fine-grained details," we were referring to what happens when patch-level features are directly used as system states, rather than stating generally that DINOv2 features themselves lack fine-grained details. Indeed, in such setups, dynamics models tend to over-focus on dominant background and robot patches, minimizing signals from small objects (e.g., cubes) that occupy very few patches, effectively omitting them from the dynamics model's predictions. We agree that DINOv2 patch-level features must contain information about small objects for the object-centric encoder to extract them. To avoid confusion, we have removed claims about addressing "omitted fine-grained details" from the updated manuscript and reworded the text for better clarity.
>
> Transitioning to an object-centric representation forces the dynamics model to weight all scene entities equitably and enables sequential planning per object during evaluation. We have revised the manuscript to accurately reflect that our performance gains stem from object decomposition for state prediction and entity-focused planning. Specifically, we updated the introduction and related work to clarify these points, and further clarified Eq. 1 in the revised text.
>
> **Concern 2: "Both Task Average" values in Table 2**
>
> **Response:** Thank you for pointing this out. The "Both Task Average" in Table 2 is the average of the success rates on Cube-Single and the average success rate on Scene-Single-Direct. Some values in this column were off by a few points as they were carried over from a prior draft. We apologize for the confusion. The overall takeaways remain unchanged, and we have updated the table to reflect the correct averages along with mean and standard deviation statistics in the revised manuscript.
>
> **Concern 3: Most components, including Slot Attention, CDiT, Diffusion Policy, Particle Filter planning, and SAM-guided object representations—already exist in prior work. The main contribution is combining these existing components into a unified framework instead of introducing fundamentally new algorithms.**
>
> **Response:** WorldDP is the first hierarchical latent-space (JEPA) world model where subgoals are generated in the top hierarchy via particle-filter-based planning on object-centric representations, and reached via a goal-conditioned diffusion policy in the lower hierarchy. While latent-space world models (e.g., Zhou et al., 2024), object-centric representations (e.g., Singh et al., 2025), and Diffusion Policy (Chi et al., 2025) are existing individual paradigms, integrating them into a unified framework for multi-stage tasks requires several novel framework-level contributions:
> - Object-Centric Representation: Existing object-centric encoders that learn over patch-level representations of backbones like DINOv2, often merge small objects with the robot arm into a single entity. We resolve this by adding a segmentation loss using SAM2-generated masks. Without this, the object-centric representation of our states would be inaccurate.
> - Dynamics Predictor: While our CDiT predictor architecture is similar to Goswami et al. (2025), WorldDP operates directly on object-centric embeddings and includes an action encoder to handle dense actions over long horizons.
> - Particle Filter (PF) Planning: Prior methods (e.g., Zhou et al., 2024) rely on CEM, which uses a single Gaussian distribution and struggles with multi-modal action distributions. We replace CEM with a PF to maintain diverse hypotheses and plan sequentially over object embeddings rather than image patches.
> - Contact Predictor: We use a learned contact predictor to explicitly guide the planning algorithm toward critical manipulation subgoals.
>
> As shown via our ablation studies (where we individually remove these components), a naive integration of these components fails to yield high performance. Instead, these specific technical modifications enable WorldDP to outperform existing world models and diffusion policies on multi-stage tasks (Tables 1 and 2). These distinctions have been made clear in Section 3 of the manuscript.

---

> > ### Author Response · Authors · 2026-07-21
> > **Response to Reviewer Sxfb Part 2/n**
> >
> > **Concern 4: The framework assumes a fixed number of object slots for each environment. It is unclear how the method would generalize to scenes containing unknown or varying numbers of objects.**
> >
> > **Response:** While our current framework assumes the number of objects is known beforehand, it can adapt to varying entity counts by setting a sufficiently large slot limit, N. During planning, the model can then focus exclusively on the slots exhibiting the most significant changes between the start and goal frames. We have added a brief discussion of this in Section 5.
> >
> > **Concern 5: The paper mainly demonstrates downstream task improvements but does not directly evaluate whether the learned object-centric representations produce more accurate latent dynamics or better long-horizon predictions than patch-level representations.
> > There is no quantitative evaluation of rollout quality, long-horizon prediction accuracy, or latent representation fidelity, making it difficult to verify whether the proposed world model is fundamentally better or whether the gains primarily arise from the hierarchical execution framework.**
> >
> > **Response:** Since each world model utilizes a distinct state space (LeWM uses a ViT [CLS] token, DinoWM uses frozen DINOv2 patch features, HWM uses downsampled patches, and WorldDP uses object-centric embeddings), a direct, universal quantitative comparison in latent space is infeasible. Furthermore, decoding these latent states back to pixel space can introduce additional artifacts, making image-based evaluation unreliable. Therefore, downstream task success rates serve as the primary proxy for dynamics accuracy. Moreover, visualization of rollout of WorldDP in the appendix shows our prediction accuracy qualitatively.
> >
> > **Concern 6: The paper claims that the proposed framework is suitable for multi-stage robotic manipulation, but the evaluation is limited to a single benchmark family (OGBench).**
> >
> > **Response:** We evaluate WorldDP on four OGBench tasks specifically selected to test challenging multi-stage skills, such as rearranging multiple cubes and executing composite actions (e.g., pressing a button followed by manipulating drawers or windows). OGBench is a standard benchmark in recent literature (e.g., Haramati et al., 2026, Maes et al., 2026b) and tests diverse skills, including pick-and-place, articulated object manipulation (e.g., drawers/windows), and physical reasoning, posing significant challenges to existing baselines such as DinoWM, LeWM, HWM, and HECRL\*. To further demonstrate the robustness of our framework, we added new experiments evaluating generalization under changes in object colors, backgrounds, camera viewpoints, and lighting, where WorldDP outperforms the established baselines in most conditions.
> >
> > Table: Generalization to unseen environment conditions: Camera Viewpoint Perturbations; Random Floor Color; Random Lighting; and Random Object Color.
> > | Task Setting | Method | Default | Camera | Floor | Lighting | Object |
> > | :--- | :--- | :---: | :---: | :---: | :---: | :---: |
> > | **Cube-Triple** | DP40 | 0.0 ± 0.0 | 0.0 ± 0.0 | 0.0 ± 0.0 | 0.0 ± 0.0 | 0.0 ± 0.0 |
> > | | DP100 | 0.0 ± 0.0 | 3.3 ± 5.8 | 0.0 ± 0.0 | 0.0 ± 0.0 | 0.0 ± 0.0 |
> > | | HECRL$^*$ | 0.0 ± 0.0 | 10.0 ± 10.0 | 3.3 ± 5.8 | 13.3 ± 15.3 | **3.3 ± 5.8** |
> > | | **WorldDP (Ours)** | **20.0 ± 17.3** | **13.3 ± 15.3** | **6.7 ± 5.8** | **30.0 ± 10.0** | 0.0 ± 0.0 |
> > | | | | | | | |
> > | **Scene-Single-Composite** | DP40 | 0.0 ± 0.0 | 0.0 ± 0.0 | 0.0 ± 0.0 | 0.0 ± 0.0 | 0.0 ± 0.0 |
> > | | DP100 | 10.0 ± 17.3 | 0.0 ± 0.0 | **3.3 ± 5.8** | 6.7 ± 11.6 | 3.3 ± 5.8 |
> > | | HECRL$^*$ | **40.0 ± 0.0** | **3.3 ± 5.8** | 0.0 ± 0.0 | 10.0 ± 10.0 | 10.0 ± 10.0 |
> > | | **WorldDP (Ours)** | 10.0 ± 10.0 | **3.3 ± 5.8** | 0.0 ± 0.0 | **16.7 ± 20.8** | **20.0 ± 10.0** |
> > | | | | | | | |
> > | **Cube-Single** | DP40 | 0.0 ± 0.0 | 0.0 ± 0.0 | 0.0 ± 0.0 | 0.0 ± 0.0 | 0.0 ± 0.0 |
> > | | DP100 | **100.0 ± 0.0** | **80.0 ± 10.0** | **83.3 ± 5.8** | **100.0 ± 0.0** | **100.0 ± 0.0** |
> > | | HECRL$^*$ | **100.0 ± 0.0** | **80.0 ± 10.0** | 23.3 ± 15.3 | 93.3 ± 5.8 | 53.3 ± 25.2 |
> > | | **WorldDP (Ours)** | 83.3 ± 5.8 | 66.7 ± 15.3 | 46.7 ± 11.6 | 76.7 ± 23.1 | 76.7 ± 5.8 |
> > | | | | | | | |
> > | **Scene-Single-Direct** | DP40 | 40.0 ± 10.0 | 13.3 ± 5.8 | 3.3 ± 5.8 | 40.0 ± 10.0 | 13.3 ± 5.8 |
> > | | DP100 | 36.7 ± 5.8 | 20.0 ± 17.3 | 0.0 ± 0.0 | 36.7 ± 5.8 | 10.0 ± 0.0 |
> > | | HECRL$^*$ | 40.0 ± 0.0 | 16.7 ± 20.8 | 10.0 ± 10.0 | 23.3 ± 5.8 | 20.0 ± 17.3 |
> > | | **WorldDP (Ours)** | **66.7 ± 15.3** | **50.0 ± 10.0** | **40.0 ± 17.3** | **66.7 ± 20.8** | **63.3 ± 5.8** |

---

> > > ### Author Response · Authors · 2026-07-21
> > > **Response to Reviewer Sxfb Part 3/n**
> > >
> > > **Concern 7: HECRL is modified by replacing its original low-level RL controller with the authors' diffusion policy. Since the baseline no longer matches the original algorithm, it becomes difficult to attribute performance differences solely to the proposed method. Additional discussion regarding fairness would strengthen the experimental comparison.**
> > >
> > > **Response:** The primary idea behind HECRL is to generate subgoals using diffusion, given start and goal frames, with the subgoals being reached by a goal-conditioned low-level RL controller. Therefore, in our experiments, for a direct comparison, we use the same HECRL framework for subgoal generation but replace their goal-conditioned RL controller with our goal-conditioned diffusion policy which takes the same inputs and outputs low-level actions. We have added a brief discussion on this in the appendix of the paper. Additionally, we have included another hierarchical baseline, HWM, in Tables 1 and 2 of the updated paper for a more comprehensive evaluation.
> > >
> > > **Concern 8: Section 3.4 details that a depth-camera-mounted pose correction is utilized during execution to handle grasp errors. It is unclear if the baselines (such as HECRL or DP100) were also granted this auxiliary correction mechanism. If they were not, the performance gains cannot be cleanly attributed to the WorldDP hierarchical framework.**
> > >
> > > **Response:** The end-effector depth-camera pose correction is integrated directly into the low-level Diffusion Policy pipeline for the Cube-Single and Cube-Triple tasks. Consequently, the Diffusion Policy baselines (DP40, DP100) and HECRL\* (which uses Diffusion Policy in its lower hierarchy) receive this exact same correction.

---

### Author Response · Authors · 2026-07-21

We sincerely thank all the reviewers and the Action Editor for their detailed feedback and time dedicated towards our work.
We appreciate their recognition of our paper's core strengths, including the **well-motivated and clearly explained algorithm** (Reviewers xosu, 8fwb, bkjx), **reasonable hierarchical design** (Reviewers xosu, Sxfb), **strong empirical results** (Reviewer xosu, 8fwb, Sxfb), **useful ablations** (Reviewers xosu, 8fwb, bkjx, Sxfb), and **adequate implementation details** (Reviewer 8fwb).

We have carefully addressed each of the concerns (weaknesses and requested changes) raised by the reviewers. As per their feedback, we have also included the following new quantitative results:
- **Statistical Rigor:** All experimental results throughout the paper have been updated to report the mean and standard deviation across seeds.
- **New Hierarchical Baselines:** Included comparisons with additional hierarchical baselines, HWM and HWM$^\dagger$, in Tables 1 and 2.
- **Generalization Experiments:** Added new generalization experiments across randomized environment conditions (Table 4), demonstrating that WorldDP outperforms baselines under almost every new condition.
- **Execution Time:** Reported planning and execution times for all methods in Table 7.

Moreover, we have clarified other conceptual concerns including ones related to the paper's contribution, motivation for using object-centric representations, and the privileged information utilized across methods (summarized in Table 8).

We hope these new experiments along with the point by point responses to the reviewers’ concerns and questions help strengthen our paper. We have incorporated the additional experiments and discussions based on the reviewers’ comments in the revised manuscript.